# Non-canonical role for *Lpar1-EGFP* subplate neurons in early postnatal mouse somatosensory cortex

**Filippo Ghezzi[1†], Andre Marques-Smith[1†§], Paul G Anastasiades[1‡#], Daniel Lyngholm[1‡], Cristiana Vagnoni[1], Alexandra Rowett[1], Gokul Parameswaran[1], Anna Hoerder-Suabedissen[1], Yasushi Nakagawa[2], Zoltan Molnar[1], Simon JB Butt[1†*]**

[1]Department of Physiology, Anatomy and Genetics, Sherrington Building, University of Oxford, Oxford, United Kingdom; [2]Department of Neuroscience, University of Minnesota, Minneapolis, United States

**\*For correspondence:**
Corresponding authorsimon.
butt@dpag.ox.ac.uk

[†]These authors contributed equally to this work
[‡]These authors also contributed equally to this work

**Present address:** [§]CoMind Technologies, London, United Kingdom; [#]Department of Translational Health Sciences, University of Bristol, Bristol, United Kingdom

**Competing interests:** The authors declare that no competing interests exist.

**Abstract** Subplate neurons (SPNs) are thought to play a role in nascent sensory processing in neocortex. To better understand how heterogeneity within this population relates to emergent function, we investigated the synaptic connectivity of *Lpar1-EGFP* SPNs through the first postnatal week in whisker somatosensory cortex (S1BF). These SPNs comprise of two morphological subtypes: fusiform SPNs with local axons and pyramidal SPNs with axons that extend through the marginal zone. The former receive translaminar synaptic input up until the emergence of the whisker barrels, a timepoint coincident with significant cell death. In contrast, pyramidal SPNs receive local input from the subplate at early ages but then – during the later time window – acquire input from overlying cortex. Combined electrical and optogenetic activation of thalamic afferents identified that *Lpar1-EGFP* SPNs receive sparse thalamic innervation. These data reveal components of the postnatal network that interpret sparse thalamic input to direct the emergent columnar structure of S1BF.

## Introduction

The emergence of function in the developing mammalian cerebral cortex is dependent on a diverse range of genetic and physiological processes that sculpt emergent network architecture. Fundamental research in animal models has revealed that transient neuronal circuits, observed in a restricted time window during early postnatal development, are a common feature of many cortical areas (*Kanold and Luhmann, 2010*; *Marques-Smith et al., 2016*). One of the first such transient circuits to be identified was that between subplate neurons (SPNs) and thalamo-recipient spiny stellate cells in layer (L)4, a circuit demonstrated to play a role in the maturation of thalamocortical synapses (*Kanold and Luhmann, 2010*; *Kanold et al., 2003*; *Tolner et al., 2012*). The subplate is a transient layer in the developing neocortex located between the emergent cortical plate and the underlying white matter (*Kostovic and Rakic, 1990*; *Hoerder-Suabedissen et al., 2015*). It contains a diverse population of neuronal subtypes that differ in term of molecular markers (*Hoerder-Suabedissen and Molnár, 2013*), morphology (*Marx and Feldmeyer, 2013*), neurotransmitter identity (*Boon et al., 2019*), and connectivity (*Viswanathan et al., 2012*). Electrophysiological studies performed in primary sensory areas suggest that SPNs are relatively mature when compared to cortical neurons in the more superficial cortical plate early in development (*Marx and Feldmeyer, 2013*; *Luhmann et al., 2000*). As such they are regarded as key mediators of early spontaneous and sensory-evoked activity (*Tolner et al., 2012*) and direct circuit maturation (*Kanold and Luhmann, 2010*). While a large proportion of SPNs undergo programmed cell death during the first postnatal

week in the mouse cortex (*Hoerder-Suabedissen and Molnár, 2013*), the surviving SPNs form a thin, compact structure below L6, termed L6b in mature neocortex (*Marx et al., 2017*; *Zolnik et al., 2020*).

The canonical role of SPNs is to support the establishment of thalamocortical synapses onto L4 neurons (*Kanold and Luhmann, 2010*; *Kanold et al., 2003*). In support of this model, previous studies have reported that SPNs receive thalamocortical input prior to innervation of L4 neurons (*Kanold et al., 2003*; *Friauf and Shatz, 1991*; *Zhao et al., 2009*; *Molnár et al., 2003*; *Higashi et al., 2002*). In turn, SPNs are proposed to form feed-forward connections onto thalamo-recipient L4 neurons in a transient circuit that is eliminated upon establishment of the mature thalamocortical connectivity in L4 (*Viswanathan et al., 2012*; *Hanganu et al., 2002*). It was proposed that, by relaying thalamic inputs to L4 through SPNs, this connectivity pattern supports developmental plasticity mechanisms (*Kanold and Shatz, 2006*) prior to the appearance of the definitive cortical architecture in primary sensory areas, e.g. ocular dominance in primary visual cortex (V1) (*Kanold et al., 2003*), barrel field formation in primary somatosensory cortex (S1BF) (*Tolner et al., 2012*), and tonotopic organisation in the primary auditory cortex (A1) (*Wess et al., 2017*). In parallel, SPNs also promote the maturation of cortical GABAergic neurons (*Kanold and Shatz, 2006*), pioneer cortico-thalamic projections (*McConnell et al., 1989*), secrete proteins that control extracellular matrix composition, attract and guide thalamocortical fibres, regulate plasticity and myelination (*Kondo et al., 2015*), and control the radial migration of cortical neurons at embryonic ages (*Ohtaka-Maruyama et al., 2018*).

However, a number of unresolved questions remain regarding SPN function in neonatal cortex: first, it remains unclear how physiological, morphological, and molecular heterogeneity of SPNs contributes to these various roles. While previous studies performed in A1 have identified two distinct physiological populations of SPN – those that receive feedback glutamatergic input from L4 and a second cohort that only receives local input (*Viswanathan et al., 2012*) – this has not been explicitly linked to SPN identity per se. To this end, we have focused on a specific, genetically defined SPN population – labeled by the *Lpar1-EGFP* transgene (*Hoerder-Suabedissen and Molnár, 2013*) to understand to what extent this population represents a homogeneous subtype of SPN and better resolve the role of these cells in neonatal somatosensory cortex. Moreover, recent evidence suggests that while thalamic input is a determinant of columnar organization in late embryonic somatosensory cortex (*Antón-Bolaños et al., 2019*), such activity pre-dates the transition to the mature cytoarchitecture and columnar signalling unit (*Dupont et al., 2006*). We sought to understand the role that *Lpar1-EGFP* SPN circuits have in interpreting such information through the first week of postnatal life up until the end of the layer 4 critical period for plasticity (CPP) at around postnatal day (P)8 in the barrel field of mouse primary somatosensory cortex (S1BF). We demonstrated that *Lpar1-EGFP* SPNs represent two distinct subtypes: (1) transient (<P5) fusiform SPNs that receive columnar input from the more superficial cortical plate but whose axons and therefore output are restricted to the SP syncytium, and (2) pyramidal SPNs that are found throughout the time period recorded (≤P8), which only receive local input from the SP network prior to P5 but whose axons traverse the full extent of the cortical plate to ramify extensively through the marginal zone. Finally, we identify that thalamic input onto *Lpar1-EGFP* SPNs in S1BF is sparse throughout early postnatal life. We propose that fusiform *Lpar1-EGFP* SPNs are ideally placed to interpret and amplify sparse thalamic input alongside emergent signalling from the cortical plate, thereby providing a template – through their innervation of other SPNs including the *Lpar1-EGFP* pyramidal subtype – for the columnar circuit assembly up until the emergence of whisker barrels in L4 at ~P4–P5. Our data suggest that *Lpar1-EGFP* SPNs do not adhere to the canonical role reported for SPNs in primary sensory cortex, and support the idea that SPNs have a variety of ways of assisting cortical circuit construction.

## Results

### Intrinsic electrophysiological and morphological diversity of *Lpar1-EGFP* SPNs

*Lpar1-EGFP* SPNs form a layer of two to three cells deep adjacent to the white matter tract in neonatal S1BF (*Figure 1a*). As a first pass to understanding the contribution of these neurons to neonatal circuits of S1BF, we recorded the intrinsic electrophysiological profiles of 103 SPNs from

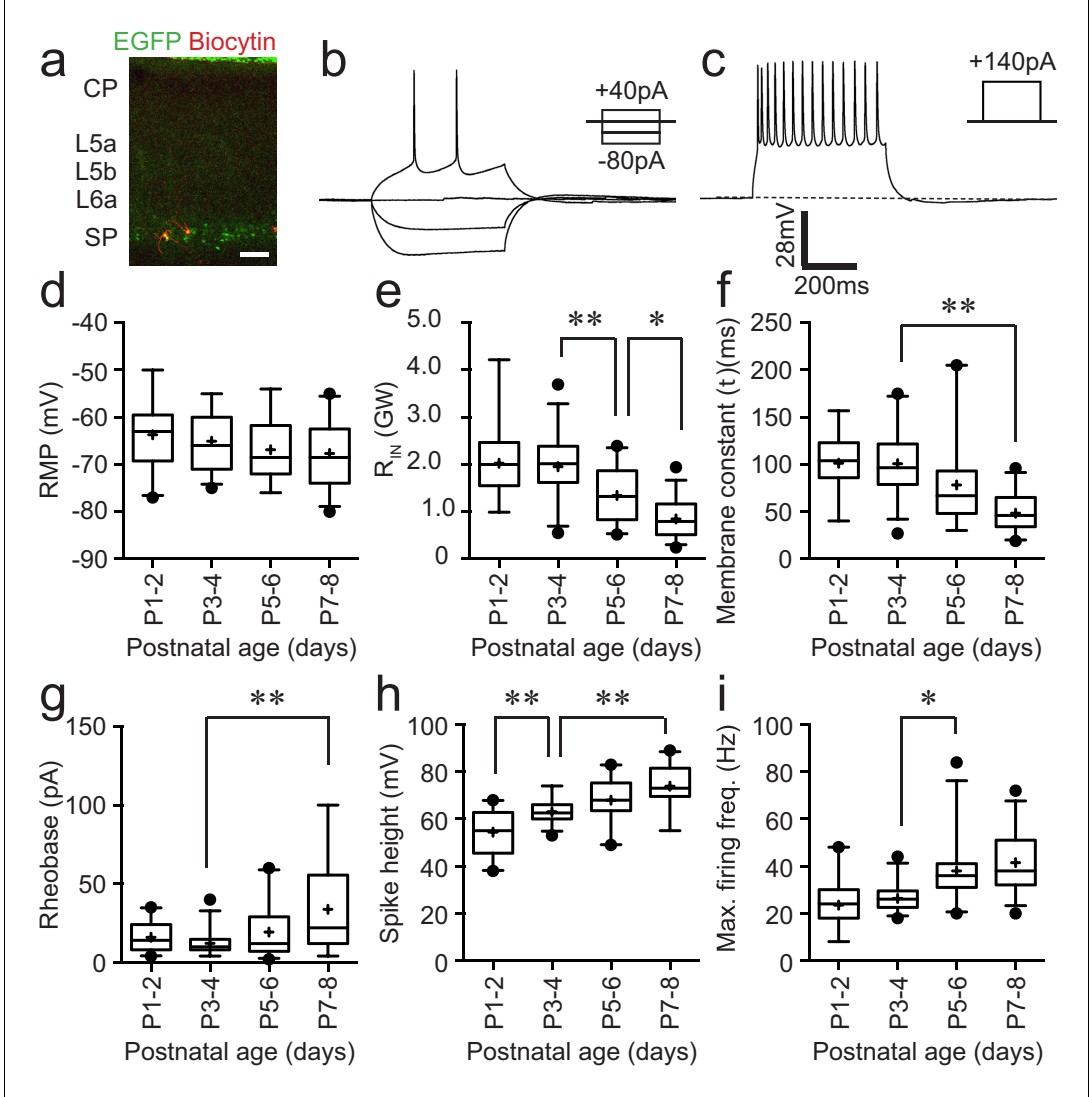

**Figure 1.** Intrinsic electrophysiological properties of Lpar1-EGFP SPNs. (**a**) Streptavidin (568 nm) labelling of record Lpar1-EGFP SPNs in mouse S1BF at P2. (**b**) Superimposed electrophysiology traces recorded from one of the cells shown in (**a**) in response to hyperpolarising and depolarising threshold current injection. (**c**) Maximum firing frequency for the same cell; scale bar is the same for (**b**) and (**c**). (**d–i**) Summary data for 103 cells grouped according to age: P1–2 (n = 22); P3–4 (n = 27); P5–6 (n = 24); P7–8 (n = 30). The range of passive membrane properties recorded included (**d**) resting membrane potential (RMP) (mV); (**e**) Input resistance ($R_{IN}$) (GΩ); (**f**) membrane time constant (tau) (ms). Active properties included (**g**) current injection required for threshold spike (rheobase) (mV); (**h**) spike amplitude (mV); (**i**) maximum firing frequency (Hz). We observed a statistically significant difference between age groups for $R_{IN}$ (ANOVA, F = 21.17, p<0.001), decay time constant (Kruskal–Wallis test, KW = 37.75, p<0.001), rheobase (Kruskal–Wallis test, KW = 15.74, p<0.01), spike amplitude (ANOVA, F = 22.05, p<0.001), and maximum firing frequency (Kruskal–Wallis test, KW = 43.15, p<0.001), whereas no statistical significant difference was observed for resting membrane potential (ANOVA, F = 1.74, p>0.05). Significant multiple comparisons of note are indicted in the relevant panels: *0.01<p<0.05; **p<0.01.

postnatal day (P) 1–8. We established that SPNs had an intrinsic electrophysiological profile consistent with regular firing pyramidal cells (***Figure 1b,c***) by injecting both depolarising and hyperpolarising current steps (500 ms) of increasing amplitude into cells recorded at resting membrane potential in current clamp configuration. Analysis of passive (***Figure 1d–f***) and active (***Figure 1g–i***) properties revealed a progressive maturation of intrinsic properties across the ages tested broadly in line with previous reports (***Hanganu et al., 2002***). Specifically, we observed a statistically significant decrease in input resistance ($R_{IN}$) (***Figure 1e***) and membrane time constant (***Figure 1f***), as well as increase in rheobase (***Figure 1g***), spike amplitude (***Figure 1h***), and maximum firing frequency (***Figure 1i***) over development. With a number of properties – membrane time constant (tau; ***Figure 1f***), rheobase

(pA; *Figure 1g*) and maximum firing frequency (Hz; *Figure 1i*), there was increased variability (± SD) with age (typically P5 onward) that suggests that not all SPNs mature at the same rate as development progressed.

Inclusion of Biocytin in the intracellular solution allowed us to reveal the morphologies of recorded SPNs (*Figure 2a*); both those assessed for intrinsic electrophysiological profile and subsequent optical stimulation experiments. In total, we recovered 58 morphologies of 103 recorded SPNs that showed complete or near complete preservation of both axonal and dendritic arbor. It was evident from our reconstruction of 19 of these cells that *Lpar1-EGFP* SPNs fell into two categories based on dendritic arbor and, specifically, the presence or absence of an apical dendrite (*Figure 2a,b*): (1) pyramidal-like SPNs with a prominent apical dendrite projecting into L6a and (2) fusiform SPNs that lacked an apical dendrite but instead exhibited bitufted dendrites that extended horizontally in the subplate and white matter tract. With the former, it was evident that the apical dendrite did not always extend perpendicular to the subplate as further revealed by analysis of the directionality of dendritic arbor (*Figure 2c*). Overlaying the axonal arbor of reconstructed morphologies revealed a further difference between these two populations: the axon of pyramidal SPNs (*Figure 2d*) ascended through the overlying cortex, with majority of cells projecting to the marginal zone/L1 (n = 9/12 cells, with the remaining three axons severed in L2). All of these cells had axonal collaterals – sometimes extensive – projecting within the subplate (SP/L6b) or adjacent L6a (*Figure 2d*). In contrast, the axon of fusiform cells was largely restricted to the SP with a few collaterals extending into L6a (*Figure 2e*) and no axon projecting to more superficial layers (n = 7/7 cells). Both cell types had extensive, but relatively simple, axonal arbors that often extended beyond the field of the low power photomicrograph either through L1 (pyramidal) or SP (fusiform). It was evident that these long-range projections extended beyond S1BF to adjacent cortical areas such as secondary somatosensory cortex (S2). Morphologies were recovered across all recorded ages; however, the proportion of fusiform cells decreased from P5 onward (*Figure 2f*). Previous analysis of the neurotransmitter phenotype of *Lpar1-EGFP* SPNs was conducted at P7 (*Hoerder-Suabedissen and Molnár, 2013*), a time point when fusiform SPNs are no longer present in our sample (*Figure 2f*). To explore the possibility that these cells represent a transient GABAergic SP population (*Qu et al., 2016*), we performed immunohistochemistry for GABA at P3 (*Figure 2g*). This confirmed that EGFP + cells in the SP were all GABA-negative (0/79 Lpar1-EGFP SPNs co-expressed GABA), while the vast majority (84%; 16/19) of EGFP+ profiles in L5 were GABA+, consistent with our previous characterisation of the SP (*Hoerder-Suabedissen and Molnár, 2013*) and L5b interneuron populations (*Marques-Smith et al., 2016*). These data identify *Lpar1-EGFP* SPNs as glutamatergic projection neurons that fall into two subtypes based on their morphology: (1) fusiform cells that innervate the SP and (2) pyramidal cells whose axons extend through the entire depth of the developing cortex to ramify extensively through the margin zone/L1 (*Figure 2h*). We found no evidence of selectively targeted axonal innervation of L4 by either subtype at the ages recorded. However, we cannot discount innervation of L4 glutamatergic spiny stellate neurons via their apical dendrites extending to L1, which transiently exist prior to the end of the first postnatal week (*Callaway and Borrell, 2011*). Finally, these data suggest that the fusiform population of *Lpar1-EGFP* SPN is a transient population of EGFP+ SPN not present in mature cortex.

## Laser scanning photostimulation reveals dynamic synaptic integration of *Lpar1-EGFP* SPNs into local glutamatergic network

We next used UV (355 nm) laser photolysis of caged glutamate to map afferent synaptic input onto SPNs in acute in vitro cortical slices (*Figure 3a*). We mapped input using laser scanning photostimulation (LSPS) across the extent of a pseudo-random (50 μm spaced) grid covering the depth of neocortex immediately above any given recorded cell. From the earliest time points recorded (P1–2) SPNs received distinct columnar glutamatergic input, either from SP and adjacent cortex (L6a) alone, or from both SP/L6a and more superficial cortex; patterns of innervation that we termed 'local' and 'translaminar' respectively (*Figure 3b*). The average laminar profile of local (n = 15) and translaminar (n = 8) SPNs at P1–2 revealed that the latter received input from the cortical plate (CP) absent in local SPNs (*Figure 3c*). This translaminar input became more prominent over the next two postnatal days (P3–4; *Figure 3d*) with the source primarily focused in the lower CP, presumptive L4. In contrast, at P5–6, we recorded relatively few translaminar neurons with the majority (84%) dominated by local SP/L6a input. The three cells defined as translaminar received afferent input from mostly

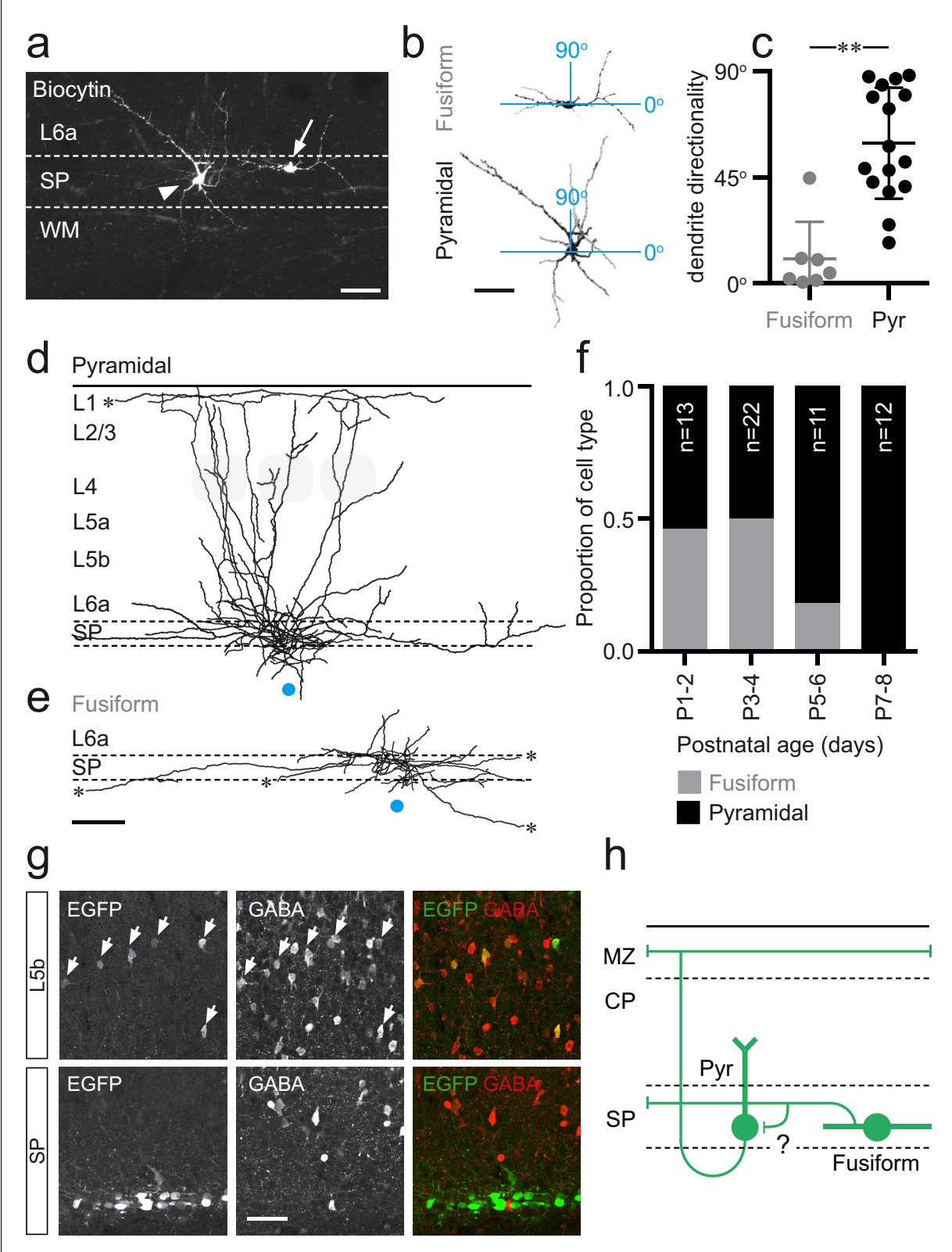

**Figure 2.** Two distinct morphologies of Lpar1-EGFP SPNs in S1BF. (**a**) Streptavidin labeled morphologies of 2 EGFP+ SPNs recorded at P3; arrowhead, pyramidal subtypes with apical dendrite projecting at ~45° into L6a; arrow, fusiform morphology with horizontal, bitufted dendrites largely restricted to the SP (scale bar = 25 μm). (**b**) Reconstructed (ImageJ) dendritic arbors of the cells shown in (**a**) were used to calculate directionality with 90° indicative of vertically orientated dendrites and ~0° primarily horizontal dendrites (scale bar = 25 μm). (**c**) A difference was observed in the dendritic orientation of

*Figure 2 continued on next page*

*Figure 2 continued*

fusiform and pyramidal SPNs (Mann–Whitney U = 5, fusiform n = 7, median = 4.38; pyramidal n = 12, median = 54.89, **p<0.001 two-tailed). Overlay of axonal arbors of (d) pyramidal (recovered between P1 and P8) and (e) fusiform (recovered P1and P5) cells aligned on soma location (horizontal position indicated by the blue circle); approximate barrel location indicated by grey-shaded areas for P5+ cells; scale bar = 180 µm. (f) Proportion of fusiform (grey) and pyramidal (black) Lpar1-EGFP SPNs over the first postnatal week. (g) Immunohistochemistry for EGFP (left) and GABA (middle); EGFP+/ GABA+ cells indicated by the white arrows; right panel, overlay of EGFP (green) and GABA (red). (h) Schematic of the two morphological subtypes of Lpar1-EGFP SPN: pyramidal (Pyr) and fusiform SPNs encountered prior to P5; a putative synaptic connection between the two subtypes is indicated by the question mark. MZ, marginal zone; CP, cortical plate; SP, suplate.

infragranular pyramidal cells (*Figure 3e*). This trend continued in the cells recorded at P7–8 although the latter were diverse in input source such that the average laminar profile and map of translaminar cells resembled a diffuse columnar band of glutamatergic input (*Figure 3f*) across L4 and L5b.

While there were clear differences in laminar input profile, the average horizontal profile for both local and translaminar SPNs did not vary through development (*Figure 3g,h*). Recovered morphologies of SPNs mapped from P1 to P4 (n = 15) revealed that all the SPNs, which received translaminar input, were of the fusiform subtype (n = 6). Two further fusiform and seven pyramidal SPNs received local input. Of the 12 SPN morphologies recovered from P5 to P8, 9 cells received local glutamatergic input. These were all of the pyramidal subtype with the exception of the only fusiform morphology recovered from ourLSPS experiments during this later window. The remaining three pyramidal SPNs received translaminar input from L5. Taken together these data suggest that transient fusiform SPNs are the primary recipients of early translaminar input from the cortical plate (*Figure 3i*), up until P4 when they become less apparent in our sample. In parallel, pyramidal SPNs are dominated by local glutamatergic input from SP/L6a at early ages (*Figure 3i*), but acquire varied translaminar input from more superficial layers from P5 onward.

## Increased cell death in subplate and adjacent cortical layers at the P4–5 transition

SPNs are regarded as a transient neuronal population, but direct evidence in support of SPN programmed cell death is limited in murine models. One possible explanation is that rapid clearance of apoptotic neurons in the developing rodent brain might preclude histological identification of dying cells. Our morphological and LSPS data point to a possible 24 hr period from P4 to P5 during a change in the make up of *Lpar1-EGFP* SPNs. To test whether this is due to cell death, we performed immunohistochemistry at P3–4 and P5–6 to assess (1) the density of EGFP+ cells, (2) expression of the apoptotic marker cleaved Caspase-3 (Casp-3+), (3) pyknotic nuclei as evidenced by DAPI staining, and (4) TUNEL staining to detect DNA breaks, in *Lpar1-EGFP* SPNs across these two time windows (*Figure 4*). Caspase-3 staining was sparse at both time points but associated with pyknotic nuclei (*Figure 4a,b*). At P3–4, Casp-3+ cells were largely restricted to the white matter (*Figure 4a*), whereas at P5–6, they were more widely distributed including in the SP. That said, we found only a couple of double-positive EGFP+/Casp-3+ SPNs (n = 2/420 cells; five animals) (*Figure 4b*). EGFP+ cells with pyknotic nuclei were however more evident, but the percentage of EGFP+ cells with pyknotic nuclei did not differ between the two time points (*Figure 4c*), despite there being a significant decrease in the density of EGFP+ SPNs (*Figure 4d*). To further examine the latter observation, we performed a TUNEL stain as an alternative means of visualizing apoptotic cells. This revealed a similar pattern of programmed cell death to our Casp-3 experiments, in that TUNEL+ cells were primarily located in the white matter at P3–4 (*Figure 4e*). However, at P5–6, we observed a significant increase in TUNEL+ profiles (*Figure 4f*) that included *Lpar1-EGFP* SPNs (*Figure 4f,g*). Moreover, it was evident that this increase was not restricted to the subplate (*Figure 4f*), so we further quantified the density of TUNEL+ profiles across the white matter, subplate, and adjacent infragranular layers at both time points, analysis that identified a surge in cell death at this later time point across the cortical layers sampled (*Figure 4h*). Overall, this suggests that while a small proportion of *Lpar1-EGFP* SPNs undergo cell death at the transition between P4 and P5, this is not specific to subplate and likely represents a wider reconfiguration of the circuit at the transition to columnar signalling in S1BF at this time (*Dupont et al., 2006*).

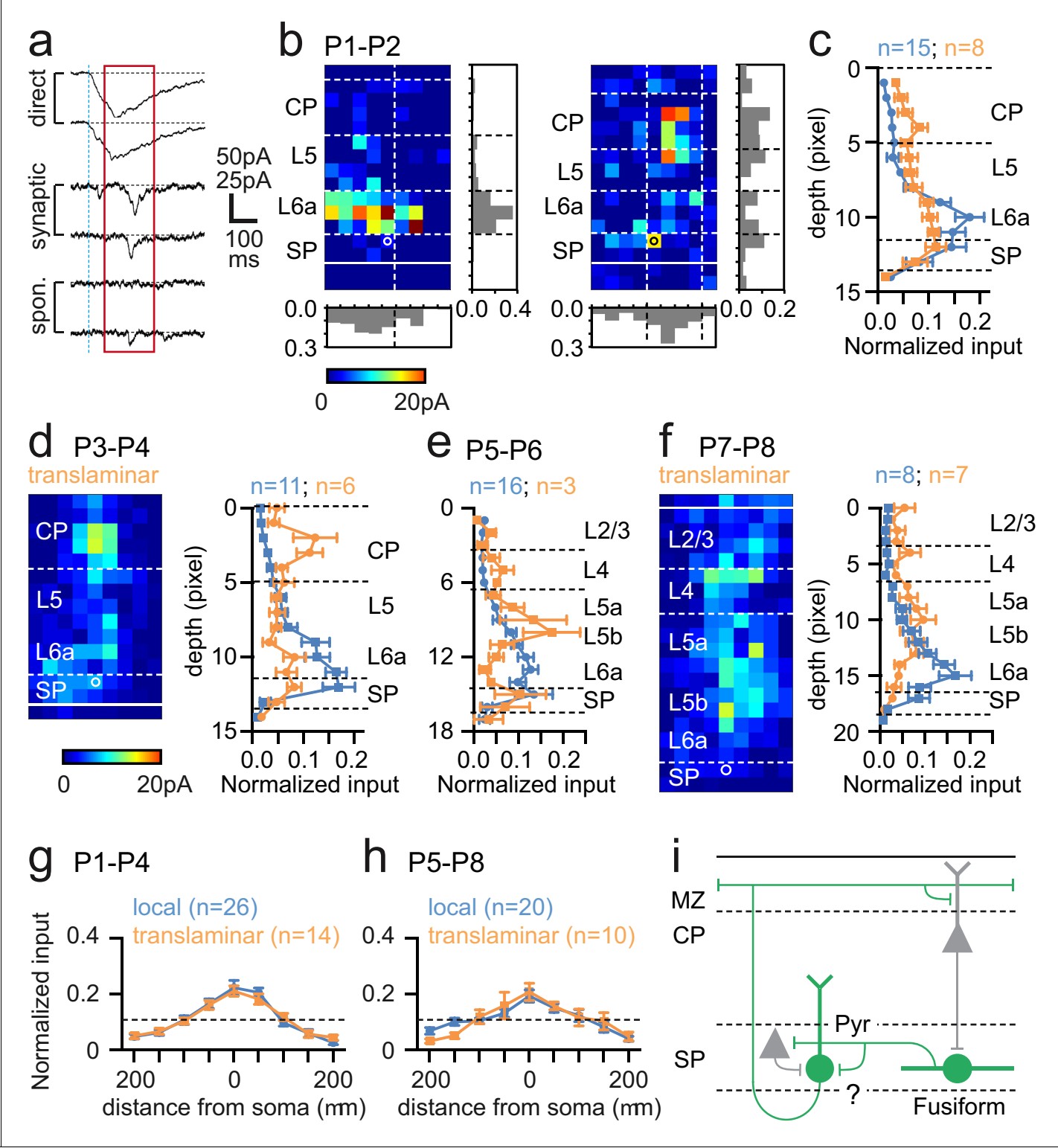

**Figure 3.** Synaptic integration of *Lpar1-EGFP* SPNs into the local cortical glutamatergic network. (**a**) LSPS of caged glutamate resulted in three responses observed in whole cell patch-clamp recordings of SPNs: top traces, large amplitude direct responses with onset locked to laser pulse onset (dashed vertical blue line); middle traces, synaptic response of consistent excitatory postsynaptic currents (EPSCs) within the monosynaptic event window (red box); bottom traces, no consistent response with occasional spontaneous EPSCs. Scale bar for direct traces: 50 pA; for synaptic and spontaneous: 25 pA. (**b**) Local (left) and translaminar (right) glutamatergic input maps for SPNs recorded at P1-2. Pixel size: 50 µm. (**c**) Average input profile for local (blue) and translaminar (orange) SPNs; translaminar SPNs showed increased input from the cortical plate (CP) and reduced local (L6a/

*Figure 3 continued on next page*

Figure 3 continued

SP) innervation. (d) Left panel, average input map for translaminar SPNs recorded at P3–4 (n=6); right panel, average input profile for local (blue) and translaminar (orange) SPNs. (e) Corresponding input profile for P5–6. (f) Average input map and profile for SPNs recorded between P7 and P8. Horizontal profile for local and translaminar SPNs aligned on cell soma at (g) P1-P4 and (h) P5–8. Horizontal axis indicates the lateral distance from the soma. (i) Schematic showing glutamatergic circuit onto *Lpar1-EGFP* SPNs.

## *Lpar1-EGFP* SPNs receive distinct sources of GABAergic input including translaminar input from somatostatin (SST+) interneurons

To understand whether the observed changes around the P4–5 transition also involve early GABAergic circuits, we repeated our LSPS experimental strategy with the cell voltage clamped at the reversal potential for glutamate ($E_{Glut}$) (*Figure 5a,b*). SPNs were pooled into two groups based on age: SPNs recorded prior to the P5 transition (P1–4; *Figure 5c–e*) and P5 onward (P5–8; *Figure 5f–h*). Similar to our previous assessment of glutamatergic input, it was evident that SPNs received either local (*Figure 5c,f*) or translaminar input (*Figure 5d,g*) across both time windows. Prior to P5, SPNs with local (*Figure 5c*) and translaminar (*Figure 5d*) input were evident in similar numbers with the latter receiving prominent columnar input from L5 (*Figure 5b,d,e*). Local GABAergic input was distributed through SP and adjacent L6a (*Figure 5c*). The average translaminar input (*Figure 5e*) revealed largely complementary distributions in GABAergic input for these two populations. From P5 onward, we observed primarily local GABAergic synaptic input onto SPNs (n = 11/16)(*Figure 5f*), with the source of translaminar GABAergic input highly variable in location resulting in a diffuse average input profile (*Figure 5g*) with a more-or-less even distribution across the depth of cortex (*Figure 5h*). Unlike glutamatergic input, the horizontal or columnar input was evenly spread for GABAergic input with the exception of translaminar input from P1 to P4 (*Figure 5i,j*). Indeed, translaminar GABAergic input onto SPNs prior to the emergence of whisker barrels at ~P5 was highly focused within the immediate column (*Figure 5i*). Recordings in the early time window yielded seven morphologies, of which four were fusiform cells that received translaminar input; the remaining three recovered neurons were the pyramidal subtype, of which two received local input (*Figure 5k*). At the later age (P5–8), we only recovered pyramidal SPN morphologies for both local and translaminar GABAergic input (n = 4).

Somatostatin (SST+) interneurons form a key component of early postnatal translaminar circuits (*Marques-Smith et al., 2016*; *Anastasiades et al., 2016*) and have been shown to drive synapse formation and circuit maturation (*Oh et al., 2016*; *Tuncdemir et al., 2016*). To test whether these interneurons (INs) influence SPNs and the circuit transition observed around P5, we first conditionally expressed Channelrhodopsin2 (ChR2) in SST+ interneurons by crossing mice homozygous for the *Ai32* (ChR2) reporter allele onto our *Lpar1-EGFP* background that was also homozygous for the *SST-Cre* driver line to generate *Lpar1-EGFP;SSTCre;Ai32* offspring. We then used wide-field blue light (470 nm) illumination to evoke SST+ IN inhibitory postsynaptic currents (IPSCs) in EGFP+ SPNs voltage clamped at $E_{Glut}$ at P3–4 (n = 7) and P5–6 (n = 5). We observed an increase in IPSC amplitude across all light powers tested greater or equal to minimal stimulation in the P5–6 when compared to the P3–4 time window (*Figure 6a*). To test whether the increase in amplitude was a result of either increased quantal size or number of innervations, we repeated these experiments in artificial cerebrospinal fluid (ACSF) in which extracellular $Ca^{2+}$ was replaced with strontium ($Sr^{2+}$). Incubation with ACSF containing $Sr^{2+}$ (Sr-ACSF) leads to asynchronous vesicular release at the presynaptic terminal providing a reasonable estimate of quantal size (*Oliet et al., 1996*; *Gil et al., 1999*). In our hands, incubation of neonatal SPNs in Sr-ACSF resulted in asynchronous release observed at minimal stimulation (*Figure 6b*) and a significant difference in ChR2-dependent IPSC amplitude between control and Sr-ACSF conditions (*Figure 6c*). However, we observed no difference in the amplitude of IPSCs recorded in Sr-ACSF between P3–4 and P5–6 time windows (*Figure 6d*) despite the significant difference in amplitude under control conditions (*Figure 6b,c*). This suggests the observed increase in amplitude at this time point results from an increase in innervation by SST+ INs rather than an increase in quantal size.

Having established that SPNs received SST+ interneuron input through the first postnatal week, we next employed conditional expression of the P2x2 receptor – an optogenetic actuator that we have previously used in conjunction with uncaging of ATP (*Anastasiades et al., 2016*) – that allows

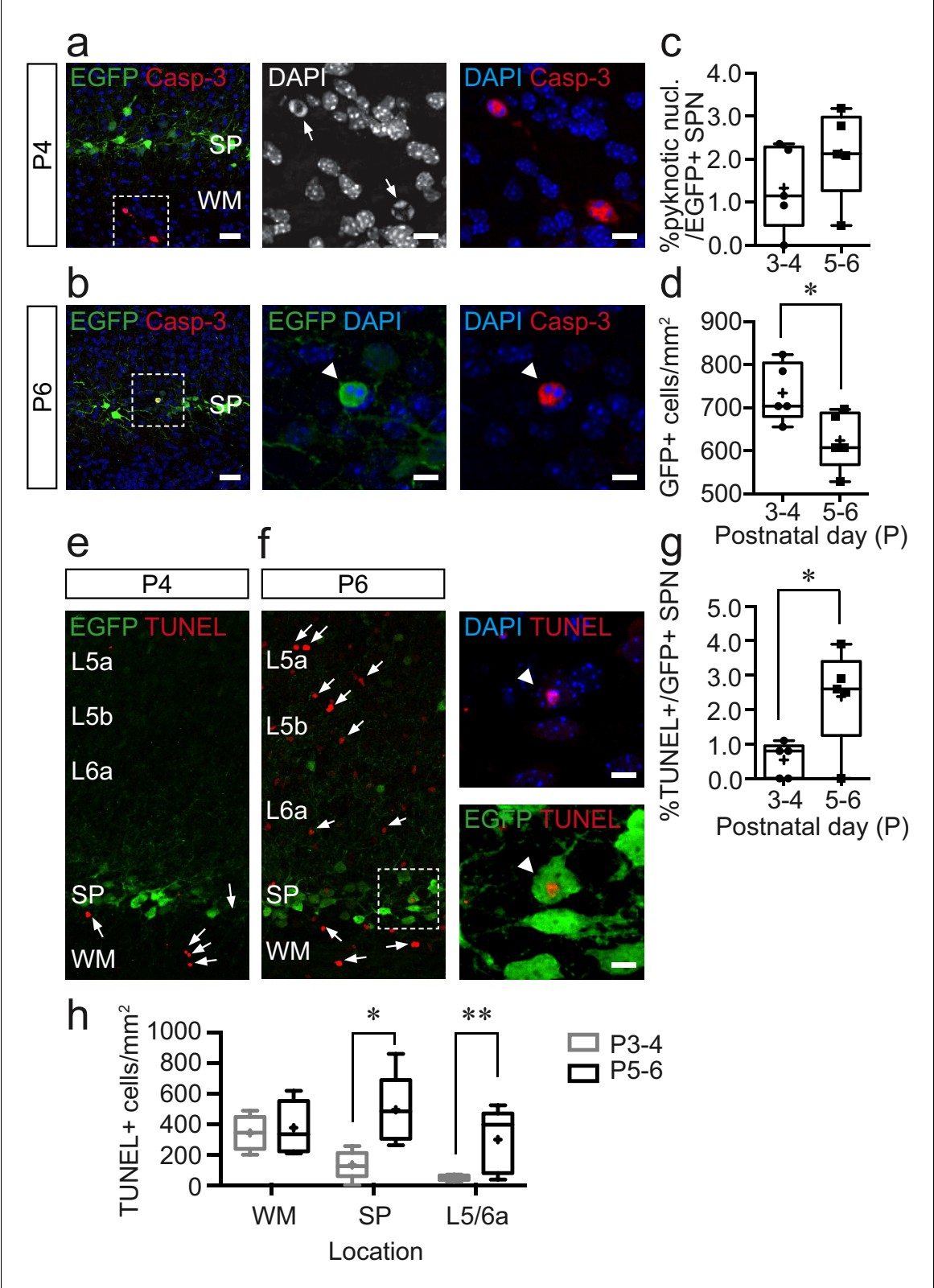

**Figure 4.** Cell death increases during the postnatal day (P)5–6 window but is not confined to the subplate. (a) Immunohistochemistry for cleaved Caspase-3 (Casp-3) and EGFP in subplate (SP) at P4; scale bar: 25 µm. Left, Casp-3+ cells in the white matter. The area bounded by the white dashed line is shown at higher magnification in the centre and right panels with Casp-3+ cells with pyknotic nuclei identified with white arrows in the centre panel; scale bar: 9 µm. (b) Similar data obtained at P6 but with white arrowhead pointing to EGFP+, Casp-3+ *Lpar1-EGFP* SPN with pyknotic nucleus.
*Figure 4 continued on next page*

*Figure 4 continued*

(c) Summary data for percentage *Lpar1-EGFP* SPNs exhibiting pyknotic nuclei at postnatal day (P)3–4 (n = 5 animals) and P5–6 (n = 5). (d) Plot showing the density of EGFP+ cells in SP at P3–4 (n = 5) and P5–6 (n = 5); asterisks, p=0.0327 (two-tailed t-test; t = 2.579, df = 8). (e,f) TUNEL staining at (e) P4 and (f) P6 with TUNEL+ cells indicated with white arrows. The area identified by the white dashed box is shown expanded in (f) in the right panels; the white arrowhead identifying a TUNEL+ *Lpar1-EGFP* SPN; scale bar: 8 μm. (g) Plot of percentage TUNEL+ *Lpar1-EGFP* SPNs at P3-4 and P5-6; asterisks, p=0.0274 (two-tailed t-test; t = 2.693, df = 8). (h) Density of TUNEL+ cells in the white matter (WM), subplate (SP), and adjacent L5/L6a across both timepoints; single asterisk, p=0.028; double asterisk, p=0.011.

us to map the somatic location of presynaptic SST+ interneurons. We performed LSPS uncaging of ATP over the 50 μm spaced pseudorandom grid to assess whether the L5 translaminar input observed from P1 to P4 originated from SST+ INs. Our analysis revealed two distinct input profiles for SPNs at this time (*Figure 6e,f*): local SP/L6a versus translaminar infragranular SST+ interneuron input. Similar to our previous findings with both glutamatergic and global GABAergic input, local input SPNs were pyramidal cells (3/3 recovered morphologies), whereas SPNs that received translaminar synaptic input were predominantly fusiform (2/3). The presence of GABAergic input onto SPNs from infragranular SST+ interneurons precedes our previously reported reciprocal connection between these SST+ cells and L4 spiny stellate neurons during the L4 CPP (P4–9) (*Marques-Smith et al., 2016*) and then onto L2/3 pyramidal cells during the emergence of L4 to L2/3 feed-forward connections (*Anastasiades et al., 2016*; *Bureau et al., 2004*). Taken together, this evidence suggests that infragranular SST+ interneurons sequentially innervate thalamo-recipient layers through early postnatal life in S1BF.

## Sparse thalamocortical input onto early postnatal *Lpar1-EGFP* SPNs in S1BF

SPNs are thought to play an important role in early thalamic integration in primary sensory cortices. To assess the role that *Lpar1-EGFP* SPNs play in the early sensory circuit we first used electrical stimulation of the ventrobasal complex (VB) of the thalamus while recording from EGFP+ SPNs in acute in vitro thalamocortical slice preparation (*Figure 7a*). Electrical stimulation evoked excitatory postsynaptic currents (EPSCs) in the majority (76%) of SPNs recorded across the time window studied, although there was a drop in incidence between the P1–4 and P5–8 (chi-square test $X^2$ (1, N = 56) = 5.364, p = 0.021) time windows (*Figure 7b*); an absence of thalamic input onto any given SPN was only recorded if TC-EPSCs were observed in other SPNs or layer four neurons in the same thalamocortical slice. We observed no antidromic potentials in recorded EGFP+ SPNs. Analysis of the amplitude of the minimal electrical stimulation EPSC (*Figure 7c*) identified a significant change in variance between these times but no difference in amplitude (p=0.08; two-tailed t-test, t = 1.92, df = 11.19). To further identify putative thalamocortical EPSCs (TC-EPSCs), we recorded the latency of the evoked EPSC, jitter (standard deviation in latency, ms), and amplitude for the minimal stimulation EPSC for 53 SPNs. All EPSCs with a latency > 10 ms and/or jitter > 1.0 ms (*Figure 7d*) were then excluded from our analysis leaving 32 SPNs that could be further divided into two populations based on 10–90% rise time and amplitude (*Figure 7e*): type 1 (n=9), large amplitude (55.3 pA ± SD 14.3), low jitter (0.20 ms ± SD 0.07), *versus* type 2 (n=23), small amplitude (14.9 mV ± SD 5.8), high jitter (0.41 ms ± SD 0.17) EPSCs (*Figure 7f*). Whether both populations represent TC-EPSCs (*Gil et al., 1999*; *Luz et al., 2017*) was unclear from our electrical stimulation of VB in part because both EPSC types conform to a previous criteria used to distinguish TC-EPSCs from antidromic cortico-thalamic EPSCs, namely that TC-EPSCs exhibit standard deviation in jitter < 1.0 ms (*Rose and Metherate, 2005*). However, thalamic connectivity could be as low as 17% if type 1 EPSCs recorded in SPNs represent true orthodromic TC-EPSCs, considerably lower than connectivity reported in previous studies of SP in early postnatal ages.

Given the disparity with previous reports, we decided to employ optogenetics in parallel with electrical simulation of VB to unequivocally identify EPSCs arising from thalamic input (*Figure 7g*). We conditionally expressed ChR2 in thalamic nuclei using the *Olig3* Cre driver line that causes recombination throughout the thalamus early in development (*Vue et al., 2007*; *Vue et al., 2009*). To validate our optogenetic strategy at early postnatal ages, we first recorded from thalamic relay neurons in *Lpar1-EGFP;Olig3 Cre;Ai32* mice and established that we could evoke (1) reliable inward currents in whole-cell patch-clamp mode in response to blue, 470 nm light (*Figure 7h*, top panel),

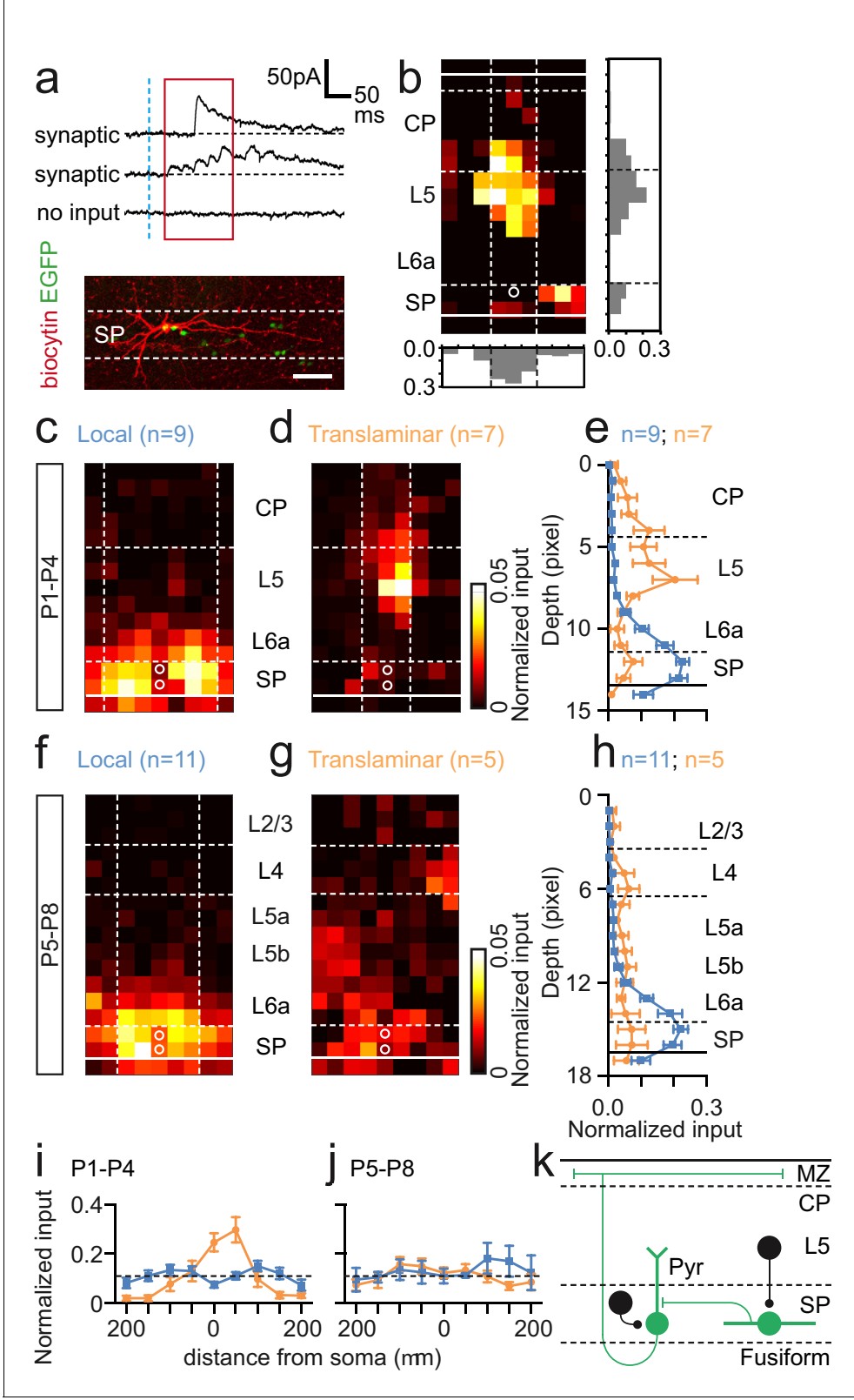

**Figure 5.** GABAergic input onto Lpar1-EGFP SPNs in the first postnatal week. (a) Top, synaptic and no input response observed in whole-cell patch-clamp recordings of SPN voltage clamped at the approximate reversal potential for glutamate (E_Glut). Bottom, recovered fusiform morphology of the SPN with translaminar GABAergic input map shown in (b). (c–e) GABAergic input onto SPNs recorded from P1–4. Average input maps for (c) local and

*Figure 5 continued on next page*

*Figure 5 continued*

(d) translaminar SPNs with profile shown in (e). (f–h) Corresponding data for SPNs recorded between P5 and P8. (i, j) Columnar analysis of GABAergic input on SPNs at (i) P1–4 and (j) P5–8. (k) Schematic of GABAergic input onto fusiform and pyramidal (Pyr) SPNs present from P1 to P4.

and (2) time-locked action potentials in cell-attached mode (*Figure 7h*, bottom panel) from the earliest time points recorded (P1) (n=5 VB cells). We then recorded *Lpar1-EGFP* SPNs and tested for TC-EPSCs using both electrical and light stimulation protocols. Across the whole time window tested (P1–8), we obtained recordings under both stimulation protocols in 19 EGFP+ cells. Of these, nine cells across all ages tested (P1–8) had short latency, low jitter (<1 ms) EPSCs in response to electrical stimulation (types 1 and 2) (*Figure 7i*). However, only two cells (11% of *Lpar1-EGFP* SPNs), with properties consistent with type 1 EPSCs, exhibited responses to both 470 nm light and electrical stimulation; no EGFP+ cells showed synaptic responses to light alone. To establish whether sparse thalamocortical connectivity is a property of the *Lpar1-EGFP* subtypes alone, we then recorded from non-EGFP+ SPNs. Using our combined electrical and optogenetic stimulation strategy, we identified thalamic input onto only 3 of 15 (27%) non-EGFP+ SPNs in the P1–4 time window. We therefore

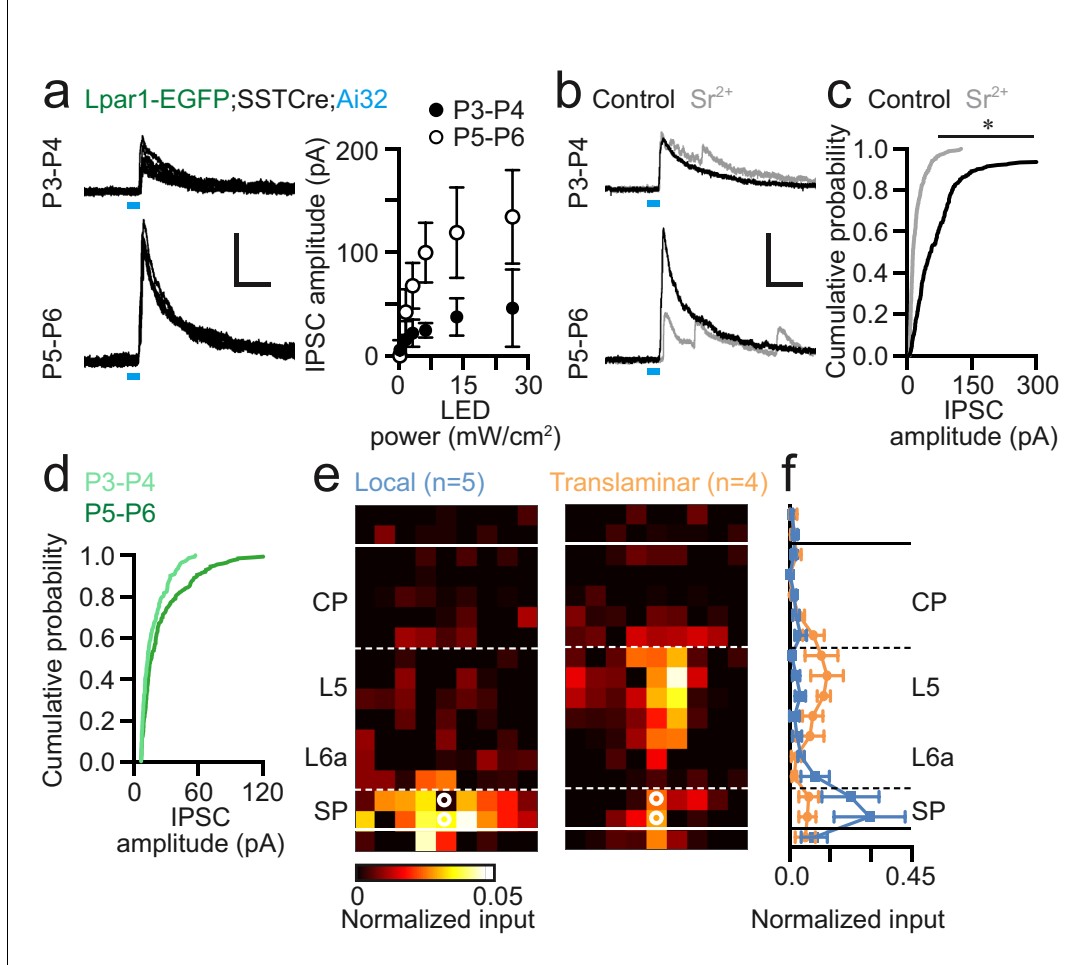

**Figure 6.** Somatostatin-positive (SST+) interneurons innervate *Lpar1-EGFP* SPNs during early postnatal life. (a) Widefield 473 nm blue light stimulation evoked IPSCs in SPNs at both P3–4 (n = 7) and P5–6 (n = 6) following conditional expression of Channelrhodopsin2 (ChR2) in SST+ interneurons. LED indicated by the blue line. (b) Incubation in $Sr^{2+}$-containing ACSF resulted in asynchronous neurotransmitter release. (c) Cumulative probability plot of ChR2-evoked IPSC amplitude for control (black line; n = 13) versus $Sr^{2+}$-containing high-divalent cation (HDC) ACSF (grey; n = 7) across P3–6; asterisk: two-sample Kolmogorov-Smirnov test: p≤0.01. (d) Comparison of early (P3–4; n=3) versus late (P5–6; n=4) IPSC amplitude in the presence of $Sr^{2+}$-containing HDC ACSF. (e) Local and translaminar SST+ interneuron input maps onto SPNs revealed through LSPS uncaging of ATP in conjunction with conditional expression of P2x2 receptor in SST+ interneurons. (f) Local (blue) and translaminar (orange) average layer profiles for SST+ input onto SPNs.

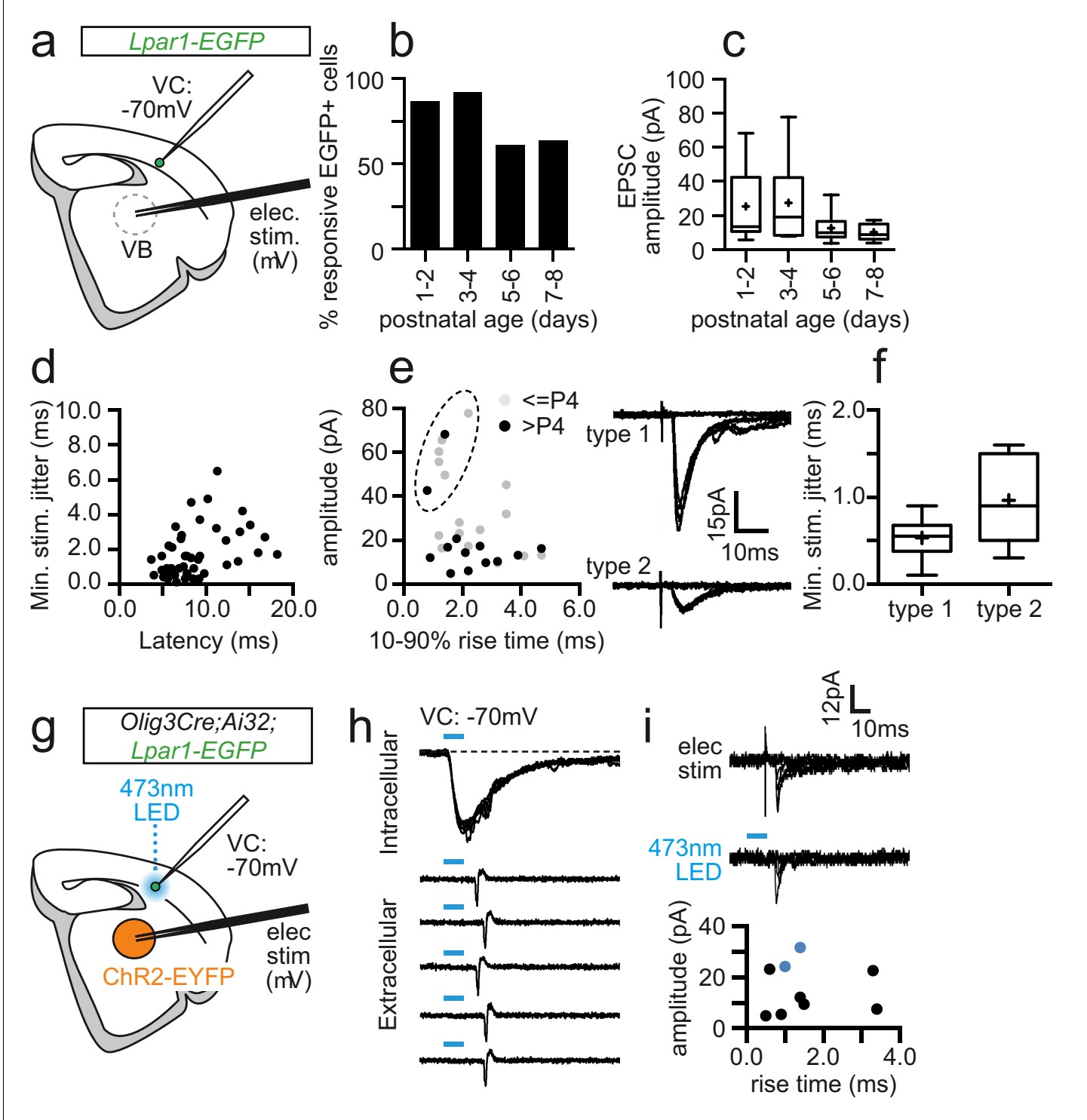

**Figure 7.** Sparse thalamic afferent input onto Lpar1-EGFP SPNs in postnatal S1BF. (**a**) Schematic showing the experimental set-up for recording electrical stimulation-evoked TC-EPSCs in thalamocortical slices. (**b**) Percentage EGFP+ SPNs exhibiting constant latency synaptic response to minimal electrical stimulation over development. (**c**) Box plot showing the average amplitude for EPSCs. (**d**) Plot of jitter (standard deviation of onset EPSC) versus average latency for all the responses shown in (**b, c**). Putative TC-ESPCs had latency ≤ 10 ms and jitter ≤ 1 ms. (**e**) Plot of average EPSC amplitude (pA) versus 10–90% rise time for all putative TC-EPSCs. Dashed circle, cluster of type 1 TC-EPSCs (top, right trace) versus type 2 (bottom trace); K-means cluster analysis (median silhouette values): K = 2; Sil = 0.89 ± 0.00; K = 3; Sil = 0.74 ± 0.04; K = 4; Sil = 0.75 ± 0.05; K = 5; Sil = 0.71 ± 0.04. (**f**) Box plot of average jitter for type 1 and type 2 EPSCs. (**g**) Schematic showing the experimental set-up for combined electrical and optogenetic

*Figure 7 continued on next page*

*Figure 7 continued*

stimulation of thalamic afferents. (**h**) (top panel) Blue light reliably evoked inward currents in thalamic relay neurons following Olig3 Cre conditional expression of ChR2, sufficient to trigger action potentials in loose cell attached recordings (bottom panel). (**i**) Electrical (top trace) and optogenetic (bottom trace) stimulation evoked TC-EPSC in a single Lpar1-EGFP SPN at P2. Bottom panel, plot of electrical stimulation evoked TC-EPSCs recorded during the combined electrical and optogenetic stimulation experiments. Blue data points indicate the two cells that also exhibited optogenetic EPSCs; dashed blue line, TC-EPSCs with a type one profile.

believe that the majority of EPSCs observed after electrical stimulation, termed type two here, arise from antidromic activation of cortico-thalamic projection neurons. Taken together these data suggest that *Lpar1-EGFP* SPNs in S1BF receive both thalamic and cortico-thalamic input in the first postnatal week, but that the former is relatively sparse, contacting only a small subset of the total population.

## Discussion

We have recorded from a genetically identified population of SPN through the first postnatal week to establish the contribution of this cell type to early circuits of somatosensory whisker barrel cortex (S1BF). Our data reveal that the *Lpar1-EGFP* SPN population comprises two distinct subtypes with different somatodendritic morphologies and afferent input throughout the first four postnatal days (P1–4): first, pyramidal SPNs that receive primarily local input at this stage from the subplate (SP) network, but whose axons traverse across all the layers of cortex to project horizontally via layer 1 (*Figure 8a*). Second, fusiform SPNs that receive translaminar input from more superficial cortical layers, but whose axons are largely confined to the immediate layer and therefore output to the SP network (*Figure 8b*). Fusiform SPNs receive glutamatergic input from the cortical plate, including putative layer 4, as well as GABAergic input from infragranular SST+ interneurons; two signalling centres that form reciprocal connections through the L4 critical period of plasticity in S1BF (*Marques-Smith et al., 2016*). During the later period (P5–8), the onset of which coincides with an increase in cell death across cortical layers, fusiform cells are encountered in significantly reduced numbers – indeed are absent in the P7–8 window. The remaining pyramidal SPNs are evenly split between those that are still dominated by local inputs and those that acquire an array of diverse inputs from across cortical layers, located both in the immediate and adjacent cortical columns. This diversification of synaptic input fits with the rapid transition to columnar signalling previously reported at P5 (*Dupont et al., 2006*) and suggests that this time point represents the switch from transient SP to layer 6b (L6b) (*Zolnik et al., 2020*) in S1BF.

Our study identifies that *Lpar1-EGFP* SPNs in S1BF have an additional novel function that does not conform to the canonical model for SPNs established across sensory cortices, wherein SPNs act as relay cells for thalamic input to L4 (*Kanold and Luhmann, 2010*; *Tolner et al., 2012*; *Allendoerfer and Shatz, 1994*). We have tested thalamic engagement with *Lpar1-EGFP* SPNs using combined electrical and optogenetic stimulation of thalamic afferent fibres and found that the incidence of connectivity onto postnatal *Lpar1-EGFP* SPNs to be as low as 11% in thalamocortical slices that otherwise showed good preservation of connectivity. This is at odds with a number of previous studies reliant exclusively on electrical stimulation (*Hanganu et al., 2002*; *Friauf et al., 1990*), which obtained levels of connectivity approaching our initial electrical stimulation paradigm. Recordings from non-*Lpar1-EGFP* SPNs suggest that this is not a property of this genetically defined population alone. As such, it is definitely worth revisiting perinatal thalamic engagement with the cortex using optogenetic approaches given that conditional expression of ChR2 in thalamic nuclei allows unequivocal discrimination of thalamocortical versus corticothalamic input without possible antidromic activation. Differences in the level of thalamic engagement aside, our data are consistent with the model that thalamic input is amplified via the subplate network (*Luhmann et al., 2009*) and onward communicated via pyramidal SPNs to more superficial layers of cortex via layer 1. Indeed, our study builds on a number of studies that advanced our understanding of the role of subplate from a simple staging post for initial thalamic innervation to a critical mediator of plasticity and amplifier of thalamic input, a role that is fulfilled by two distinct morphological subtypes. However, how this amplified signal is relayed to the overlying cortex is unclear beyond that it is likely mediated by the pyramidal subtype with axons ramifying in the marginal zone. None of our recovered morphologies

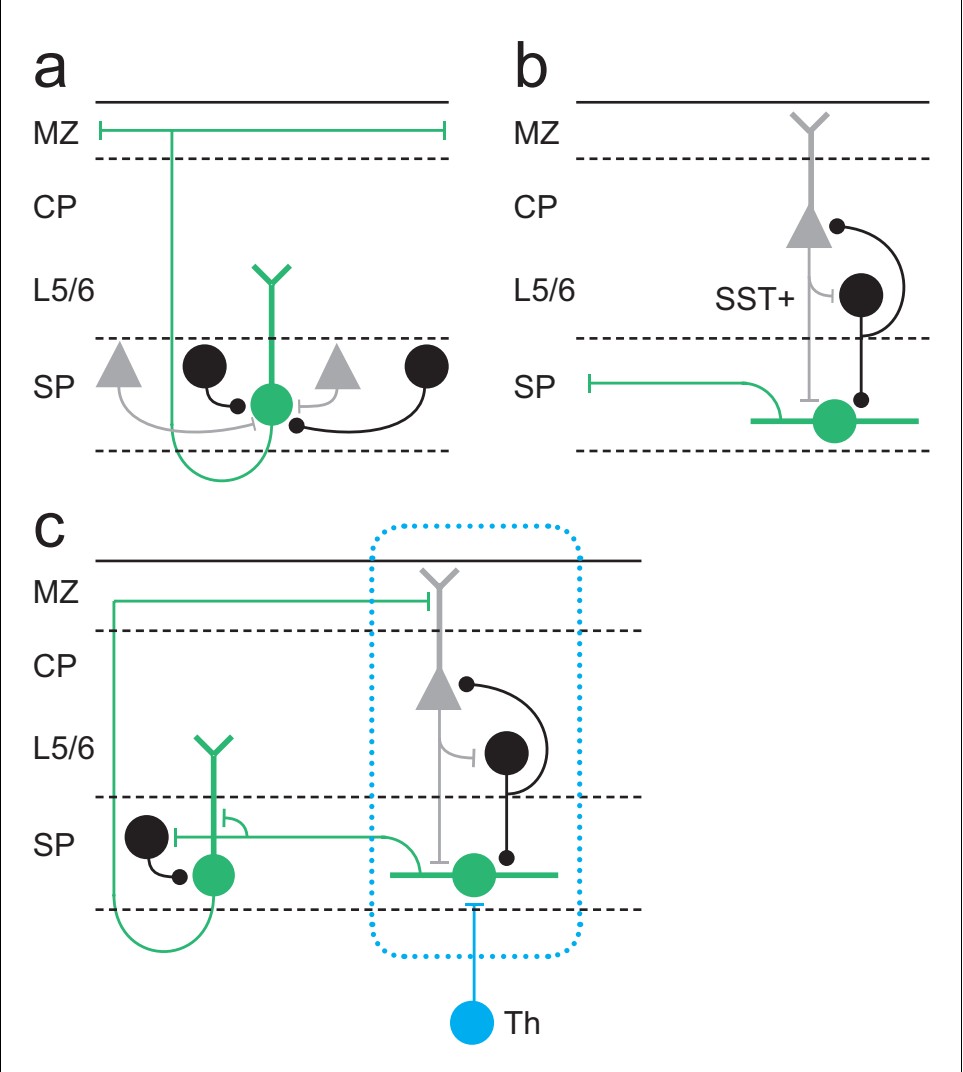

**Figure 8.** Model for Lpar1-EGFP SPN circuits in early postnatal cortex. (**a**) Pyramidal Lpar1-EGFP SPNs received broad glutamatergic (grey neurons) and GABAergic (black) input from the subplate zone. (**b**) Transient fusiform Lpar1-EGFP SPNs in contrast receive translaminar input from glutamatergic neurons (grey) in the cortical plate and SST+ interneurons (black) in infragranular layers. (**c**) We propose that sparse thalamic input (Th) onto Lpar1-EGFP SPNs interacts with both fusiform and pyramidal SPN circuits to sculpt the emergent columnar cytoarchitecture (blue dashed box). .

suggest dense innervation of L4 per se by this particular genetically defined subpopulation of SPN (*Friauf et al., 1990*). Indeed, this and our extensive mapping of S1BF using LSPS through early postnatal life (*Anastasiades et al., 2016*; *Anastasiades and Butt, 2012*) provide little evidence that there is a privileged route of connectivity between SP and L4 for a protracted period during early postnatal life in this primary sensory area, in contrast to other sensory modalities.

We targeted a genetically defined population of SPN using the *Lpar1-EGFP* transgenic mouse line (GENSAT). Lpar1 (Edg2) is one of a number of markers previously shown to delineate SPN diversity (*Hoerder-Suabedissen and Molnár, 2013*), with the cohort labeled by this line one of the earliest born subtypes with peak neurogenesis at embryonic day (E)11.5. EGFP expression is evident in this population at embryonic ages (*Hoerder-Suabedissen and Molnár, 2013*), increasing to label both SP and non-SP neurons by the end of the first postnatal week (*Marques-Smith et al., 2016*). This reported increase in strength of EGFP expression in SPNs through early postnatal life would seem to preclude the possibility that the fusiform SPN subtype down-regulate EGFP at P5. The most

parsimonious explanation – that reconciles our observation of fusiform *Lpar1-EGFP* cells as a 'classical' transient SPN population with the fact that there is no significant decrease in the number of EGFP+ SPNs over this time window (*Hoerder-Suabedissen and Molnár, 2013*) – is that this subtype represent only a small fraction of the EGFP+ SPN number. Our observed increase in the prevalence of the pyramidal subtype of EGFP+ SPNs over development has been documented by others (*Marx et al., 2017*). Finally, the *Lpar1-EGFP* transgenic line is an excellent tool for targeting and recording these two morphological variants of SPN. However, our knowledge of the efferent targets of these cells is, in absence of a conditional genetic strategy, limited to a purely morphological assessment. Alternative genetic approaches such as CRE-DOG (*Tang et al., 2015*) although useful to researchers targeting EGFP+ neuronal populations in the adult cortex (*Naka et al., 2019*) do not provide a viable means of targeting the *Lpar1-EGFP* cells within the first postnatal week. What is evident from our analysis is that the two subtypes present in the earlier time window target completely different layers within the developing cortical plate, highly suggestive of different roles within the early circuitry of S1BF.

LSPS has been used previously in conjunction with glutamate uncaging to probe the early SP circuits of primary auditory cortex (A1) (*Viswanathan et al., 2012*; *Zhao et al., 2009*). Similar to these reports, we find SPNs that receive local and translaminar input. However, the temporal dynamics of the connections that we observe in S1BF, notably the early afferent input from L4, are quite different to those reported for A1 wherein L4 input emerges in the second postnatal week (*Viswanathan et al., 2012*). This disparity in timing could underpin differences in the role of SP in circuit maturation between sensory areas and suggests that the cytoarchitecture of sensory areas differs from the outset.

In recent years, optical approaches have extended our knowledge of the early circuit of somatosensory cortex. It is evident that prenatal spontaneous thalamic activity plays an instrumental role in determining the columnar organization of S1BF (*Antón-Bolaños et al., 2019*). Shortly before birth thalamic afferent fibres are restricted to the subplate and thalamic stimulation elicits activity that spreads laterally through this and immediate adjacent infragranular layers (*Higashi et al., 2002*). It is possible that at these early ages thalamic innervation of SPNs is widespread in S1BF and that the role of such spontaneous activity is to competitively select the sparse SPNs that will maintain thalamic innervation into the first few postnatal days, a time point by which sensory activity has already transitioned to the overlying cortical layers in S1BF (*Antón-Bolaños et al., 2019*). Our data suggest that *Lpar1-EGFP* pyramidal SPNs are likely conduits for such activity, recruiting pyramidal cells and GABAergic interneurons in more superficial cortex (*Friauf and Shatz, 1991*; *Che et al., 2018*) via their L1 axon collaterals. In turn, neurons in the superficial cortical layers provide feedback columnar glutamatergic and GABAergic synaptic input onto transient fusiform SPNs thus completing the circuit (*Figure 8c*). This indirect mechanism could provide the necessary framework for the interpretation of early thalamic signals resulting in columnar organisation. Indeed, such a mechanism coupled with the sparse nature of thalamic engagement with the subplate represents a plausible substrate to ensure the emergence of spatially distinct columnar circuits in S1BF. Moreover, relay of thalamic input via L1 moves away from a L4-centric view of early thalamic engagement, more in line with distributed thalamocortical input across all six layers of neocortex (*Feldmeyer et al., 2013*). Our findings provide insight into the earliest neuronal networks of somatosensory cortex, highlighting transient glutamatergic and GABAergic circuits that are essential for the emergence of normal perception.

## Materials and methods

**Key resources table**

| Reagent type (species) or resource | Designation | Source or reference | Identifiers | Additional information |
|---|---|---|---|---|
| Strain, strain background (*Mus musculus*, male) | *Tg(Lpar1-EGFP)GX193Gsat* | GENSAT Project at Rockefeller University | MGI:4847204; transgene insertion GX193 | Y Chromosome linked |

*Continued on next page*

*Continued*

| Reagent type (species) or resource | Designation | Source or reference | Identifiers | Additional information |
|---|---|---|---|---|
| Strain, strain background (*M. musculus*, mixed sex) | $Sst^{tm2.1(cre)Zjh}/J$ | The Jackson Laboratory | Stock No: 013044; RRID:IMSR_JAX:013044 | Maintained in Butt lab on C57BL/6 background |
| Strain, strain background (*M. musculus*, mixed sex) | $Olig3^{tm1(cre)Ynka}$ | Prof. Yasushi Nakagawa University Minnesota | | |
| Strain, strain background (*M. musculus*, mixed sex) | $Gt(ROSA)26Sor^{tm32(CAG-COP4*H134R/EYFP)Hze}/J$ | The Jackson Laboratory | Stock No: 012569; RRID:IMSR_JAX:012569 | Maintained in Butt lab on C57BL/6 background |
| Strain, strain background (*M. musculus*, mixed sex) | *R26::P2x2r-EGFP* (floxed-stop-rat P2x2 receptor) | Prof. Gero Miesenböck (Oxford) | | Maintained in Butt lab on C57BL/6 background |
| Antibody | Chicken anti-GFP (polyclonal) | Abcam | ab13970 RRID:AB_300798 | (1:250) |
| Antibody | Rabbit anti-GABA (polyclonal) | Sigma-Aldrich | Cat# A2052, RRID:AB_477652 | (1:1000) |
| Antibody | Guinea-pig anti-GABA (polyclonal) | Abcam | ab17413 RRID:AB_443865 | (1:1500) |
| Antibody | Rabbit anti-Caspase three antibody, active (cleaved) form (polyclonal) | Merck Millipore | AB3623 RRID:AB_91556 | (1:200) |
| Antibody | Goat anti-Chicken, Alexa Fluor 488 (polyclonal) | Abcam | ab150169 RRID:AB_2636803 | (1:1000) |
| Antibody | Goat anti-Rabbit, Alexa Fluor 546 (polyclonal) | Thermo-Fisher | A11035 RRID:AB_143051 | (1:1000) |
| Commercial assay or kit | In Situ Cell Death Detection Kit, TMR red | Roche, Sigma-Aldrich | SKU: 12156792910; Lot #: 45197500 | |
| Chemical compound, drug | NaCl; KCl; NaHCO$_3$; NaH$_2$PO$_4$; MgCl$_2$; CaCl$_2$; K-gluconate; Li-GTP; Mg-ATP; HEPES; Gluconic acid; EGTA; SrCl$_2$ | Sigma-Aldrich | S7653 P4504 S6014 S3139 M2670 22,350–6 G4500 G5884 A9187 H3375 G1951 E4378 204463 | |
| Chemical compound, drug | MNI-caged glutamate | Tocris Bioscience UK | Cat. No.: 1490. Lot no.: 48A/206152; 50A/255070; 50A/220558; 50A225070; 51A/240237; 51A/243753; 52A/249897; 52A/251627 | 100 µM |

*Continued on next page*

*Continued*

| Reagent type (species) or resource | Designation | Source or reference | Identifiers | Additional information |
|---|---|---|---|---|
| Chemical compound, drug | DMNPE-caged ATP | Life Technologies UK | A1049 | 100 µM |
| Chemical compound, drug | Biocytin | Sigma-Aldrich | B4261; Lot no. SLCB0219; SLBS5344 | |
| Chemical compound, drug | Streptavidin, Alexa Fluor 568 conjugate | Thermo-Fisher | S11226 RRID:AB_2315774 | (1:500) |
| Software, algorithm | GraphPad Prism Software | GraphPad Software | RRID:CR_002798 | |
| Software, algorithm | ImageJ/Fiji | Fiji | RRID:SCR_002285 | |
| Software, algorithm | pClamp 10 | Molecular Devices | RRID:SCR_011323 | |
| Software, algorithm | MATLAB | Mathworks | RRID:SCR_001622 | |
| Other | DAPI stain | Thermo-Fisher | D1306 RRID:AB_2629482 | (1:1000) |

## Animal husbandry and use

Animal care and experimental procedures were approved by the University of Oxford local ethical review committee and conducted in accordance with UK Home Office personal and project (70/6767; 30/3052; P861F9BB75) licenses under the Animals (Scientific Procedures) 1986 Act. The following mouse lines were used: *Lpar1-EGFP* (Tg(Lpar1-EGFP)GX193Gsat), *SST-ires-Cre* (Sst$^{tm2.1(cre)Zjh}$/J), *Olig3 Cre* (Olig3$^{tm1(cre)Ynka}$), *Ai32* (*Gt(ROSA)26Sor$^{tm32(CAG-COP4*H134R/EYFP)Hze}$*/J), and *R26::P2x2R-EGFP* (floxed-stop-rat P2x2 receptor). All experiments were performed blind to the mouse genotype with the exception of *Lpar1-EGFP* transgene, which is Y chromosome linked (*Hoerder-Suabedissen and Molnár, 2013*). The date of birth was designated postnatal day (P)0.

## Acute in vitro slice preparation

Acute brain slices were prepared as previously described (*Marques-Smith et al., 2016*). Male mice (P1–8) were anesthetised with 4% isoflurane in 100% $O_2$ and decapitated; the cerebral cortex was quickly dissected in ice-cold, oxygenated (95% $O_2$/5% $CO_2$) ACSF of the following composition (in mM): 125 NaCl, 2.5 KCl, 25 NaHCO$_3$, 1.25 NaH$_2$PO$_4$, 1 MgCl$_2$, 2 CaCl$_2$, 20 glucose (300–310 mOsm; all chemicals were purchased from Sigma unless otherwise stated). Coronal and thalamocortical slices (350–400 µm) including the primary somatosensory barrel cortex (S1BF) were cut in ice-cold ACSF through a vibratome (Vibratome 3000 Plus; The Vibratome Company) and allowed to recover in ACSF at room temperature (RT) for at least 1 hr prior to electrophysiological recordings. Coronal slices were obtained by cutting the brain at an angle perpendicular to S1BF; thalamocortical slices were obtained according to established procedures with the angle varied according to developmental age (*Marques-Smith et al., 2016*; *Agmon and Connors, 1991*).

## Whole-cell patch-clamp electrophysiology

Slices containing S1BF were selected for electrophysiology experiments if they showed good preservation of the radial structure, as assessed by the presence of layer (L)5 pyramidal neuron apical dendrites extending to supragranular layers. *Lpar1-EGFP* SPNs were readily distinguished from *Lpar1-EGFP* GABAergic interneurons based on their localisation in a thin layer of cells located between the cortical L6 and the underlying white matter, identified as the SP. SP could be detected as a thin, compact cell layer that could be distinguished from Layer 6a and white matter. Cells were selected ~50 µm below the slice surface and targeted for patch-clamp recordings guided through infrared-differential interference contrast (IR-DIC) microscopy using a 40× water-immersion objective. Whole-cell patch-clamp electrophysiological recordings were performed at RT using a Multiclamp 700B

amplifier and Digidata 1440A digitizer (Molecular Devices). Patch pipettes were obtained from borosilicate glass microelectrodes (6–9 MΩ; Harvard Apparatus, UK), pulled through a PC-10 puller (Narishige, Japan). Electrodes were filled with either a K-based (128 mM K-gluconate, 4 mM NaCl, 0.3 mM Li-GTP, 5 mM Mg-ATP, 0.1 mM CaCl$_2$, 10 mM HEPES; pH 7.2 with KOH; 280–290 mOsm) or Cs-based intracellular solution (100 mM gluconic acid, 0.2 mM EGTA, 5 mM MgCl$_2$, 40 mM HEPES, 2 mM Mg-ATP, 0.3 mM Li-GTP; pH 7.2 with CsOH; 280–290 mOsm). Biocytin (0.3%) was included in the intracellular solution to allow the morphological reconstruction of recorded neurons. To study EPSCs, SPNs were held at a holding potential (V$_h$) of −60 mV; IPSCs were recorded by voltage-clamping the cell near the equilibrium potential for glutamate (E$_{Glut}$). For mapping of IPSC input, E$_{Glut}$ was found empirically by uncaging glutamate in the proximity of the recorded cell and tuning the V$_h$ until little or no net laser-induced direct postsynaptic current was observed. For optogenetic experiment, V$_h$ was set to 0 mV (corrected for calculated liquid junction potential of ~13 mV for the Cs-based intracellular solution). All recordings were sampled at 20 kHz and low-pass filtered online at 0.5 kHz. Cell input and series resistance (R$_{in}$ and R$_s$) were monitored throughout the duration of the recording without applying compensation; recordings were discarded when R$_s$ exceeded 20% of its initial value.

Cells patched with the K-based intracellular solution were initially held in current-clamp configuration to record their intrinsic electrophysiological profile prior to LSPS experiments. Intrinsic electrophysiological properties were assessed using both depolarising and hyperpolarising current steps (500 ms) of increasing amplitude applied from resting membrane potential; step size was adjusted based on the input resistance of the cell.

## Laser-scanning photostimulation: methods and analysis

Laser-scanning photostimulation (LSPS) was performed according to the method previously described (*Anastasiades and Butt, 2012*; *Anastasiades et al., 2018*). This optical technique allows to stimulate neurons in a small portion of cortical tissue (~50 μm) while recording the postsynaptic current in the target neurons. Thus, the location of any presynaptic neurons showing functional connectivity to the recorded one can be inferred by the location of the optical stimulation. Prior to LSPS, slices were incubated for a minimum of 6 mins. with high-divalent cation (HDC) ACSF of similar composition to the normal ACSF but with increased concentration (4 mM) of MgCl$_2$ and CaCl$_2$ and supplemented with 100 μM MNI-caged glutamate (Tocris Bioscience, UK) for glutamate uncaging experiments. We perform LSPS experiments in HDC ACSF solution to reduce polysynaptic transmission and the occurrence of spontaneous synaptic currents. Mapping of cell-type-selective inputs were performed with an optogenetic strategy previously described (*Anastasiades et al., 2016*). In brief, the P2x2 receptor was conditionally expressed into SST+ interneurons and selectively stimulated by laser uncaging of DMNPE-caged ATP (100 μM, Life Technologies, UK). LSPS was performed using an ultraviolet (UV) laser (DSPL-355/30) and a galvanometer targeting system (UGA-42, Rapp Optoelectronic GmbH, Germany) focused through a 10× Olympus objective. The stimulation grid was organised into 17 × 9 target spots (~50 μm spatial resolution). Long-duration (100 ms), low-power (<2 mW at sample plane) laser pulses were fired in a pseudo-random order at 1–2 Hz frequency. In order to cover the whole extent of the cortical column, two to three LSPS grids were sequentially employed and properly aligned and averaged offline during data analysis. For each individual LSPS grid, a minimum of three runs were obtained and averaged.

Electrophysiological current traces were analysed with Minianalysis 6.0 (Synaptosoft Inc) to extrapolate amplitude and onset time of all IPSCs or EPSCs recorded. Direct responses to glutamate were identified by their short latency and slow onset kinetics (time to peak ~100 ms) and excluded from subsequent analysis. Only EPSCs characterised by fast onset kinetics were considered for further analysis if they occurred within the putative detection window for monosynaptic events, determined according to previously published criteria (*Anastasiades and Butt, 2012*). For each laser spot of the grid, all events whose onset fell within this detection window were summed and then averaged with values from different runs of the same experiment. Final heatmaps were built through a customised Matlab (Mathworks, USA) script. In order to allow the reconstruction of the layer profile on the input map, a photomicrograph of the grids relative to the slice preparation was acquired, and layer boundaries were manually determined. Normalised heatmaps were generated by dividing the value in each spot by the sum of all pixels. Linear profiles (layer and columnar) were obtained by

summing all values for each line in individual heatmaps. Average maps were obtained by aligning each individual map to the SP/Layer 6a boundary and averaging corresponding pixels.

### *In vitro* optogenetics stimulation

Optogenetic experiments were performed by conditionally expressing Channelrhodopsin 2 (ChR2; via the *Ai32* reporter allele) in SST+ interneurons (using *SST-ires-Cre*) or thalamic relay neurons (using *Olig3 Cre*). Wide-field light stimulation was delivered through a 40× objective to focus blue (470 nm LED, CoolLED, UK) light onto the recorded cell. For each recorded SPN, two light stimulation duration pulses (1 and 10 ms) were employed at multiple LED power intensities to ensure that the minimal stimulation and full range of activation was captured irrespective of developmental age. For each LED pulse duration and intensity, five pulses were administered at a 20 s interval and the evoked postsynaptic current (PSC) recorded.

Data analysis was performed through a customised Matlab script. Light-evoked PSCs were analysed if their onset was detected within 25 ms from the onset of the light stimulus; the relatively long latency was used to account for developmental effects that may affect ChR2 expression. For events within the mono-synaptic detection window, multiple PSC features were extracted such as amplitude, latency, 10–90% rise time, and decay time constant (τ). In particular, the latency of the PSC was calculated from the onset of LED stimulation; the decay τ was found by fitting a mono-exponential curve to the decay phase of the PSC; percentage of PSC occurrence was calculated throughout the five sweeps at each LED intensity.

In a subset of experiments, recordings of light-evoked IPSCs from SST+ interneurons were performed in a modified HDC ACSF containing 4 mM $SrCl_2$ to replace $CaCl_2$. Due to the slow onset of its effects, slices were bathed in $Sr^{2+}$-containing HDC for a minimum of 20 min before light stimulation and data collection (*Gil et al., 1999*).

### Electrical stimulation of thalamic afferents

Thalamocortical (TC) afferent input to SPNs was tested using a bipolar microelectrode (Harvard Apparatus, UK) placed either in the ventrobasal nucleus (VB) of the thalamus or the internal capsule (IC) and connected to a current isolator (DS3, Digitimer Ltd, UK). The strength of the electrical stimulation was varied to find the minimal stimulation value (*Raastad et al., 1992*; *Isaac et al., 1997*), corresponding to EPSC evoked on ~50% of trials. The interstimulus interval was set at either 30 or 60 s depending on developmental age. TC-EPSCs were considered if calculated standard deviation (jitter) of the EPSC latency was <1 ms at minimal stimulation.

### Morphological reconstruction of recorded cells

Following electrophysiological assessment, slices containing biocytin-filled cells were fixed in 4% paraformaldehyde (PFA; diluted in phosphate-buffered saline, PBS) overnight at 4°C. Slices were then rinsed in PBS and incubated in 0.05% PBST containing Streptavidin-Alexa568 (1:500; Molecular Probes, USA) for 48–72 hr at 4°C. Slices were then washed 3× 10 min in PBS and mounted on histology slides with Fluoromount (Sigma) mounting medium.

Slices were imaged through an Olympus FV1200 confocal microscope equipped with 10× or 20× dry objective. Z-stack images were acquired in order to maximise imaging of all neuronal processes containing biocytin to allow the offline morphological reconstruction. Image analysis was performed with Fiji-ImageJ software (NIH): confocal images of filled cells were selected for morphological reconstruction, performed using the Simple Neurite Tracer plugin. Dendrite directionality was calculated using the Directionality plugin implemented in Fiji-ImageJ onto reconstructed dendritic morphologies.

### Immunohistochemistry and TUNEL stain

Mice were terminally anesthetised with pentobarbital (90 mg/kg) and transcardially perfused with 4% PFA in PBS. Dissected brains were incubated in PFA for 2 hr at 4°C and then cryoprotected in 20% sucrose for 24 hr at 4°C. Brains were then embedded into O.C.T. (VWR), frozen on dry ice, and stored at −80°C. Each brain was sectioned into 14–16 μm thick slices and mounted on histology slides; slides were stored at −20°C. Slides selected for immunohistochemistry were air-dried overnight at RT and washed 3× 10 min at RT in PBS. Slides were then permeabilised for 30 min in 0.5%

PBST (0.5% Triton X-100 in PBS) and incubated in blocking solution (PBST 0.1%, Normal Goat Serum 5%) for 1 hr at RT. Primary antibodies used were chicken anti-GFP (ab13970, Abcam, dilution 1:250), rabbit anti-GABA (A2052, Sigma, 1:1000), guinea-pig anti-GABA (ab17413, Abcam, 1:1500), and rabbit anti-Caspase-3 (AB3623, Merck Millipore, 1:200). Slides were incubated in primary antibody solution overnight at 4°C. Slides were then washed 3× 10 min at RT in PBS and subsequently incubated in secondary antibody (Goat anti-Chicken IgG Alexa Fluor 488 conjugate, Goat anti-Rabbit IgG Alexa Fluor 568 conjugate; diluted in blocking solution 1:1000) for 2 hr at RT. Finally, the slides were washed in PBS, counterstained with DAPI (diluted 1:1000 in PBS) for 3 min at RT, and mounted with Fluoromont (Sigma). For the TUNEL stain, slides were processed for GFP immunohistochemistry as above, and then the TUNEL stain was applied as per the supplier (Roche, SKU: 12156792910) instructions. Slides were then counterstained with DAPI and mounted as above. All slides were imaged through an Olympus FV1200 confocal microscope.

## Statistical analysis

All results are expressed as mean ± standard error of the mean; n indicates the number of cells recorded. Statistical analysis was performed with Prism (GraphPad, USA). Normality and equal variance tests were run to direct the appropriate statistical test choice for comparison of parametric versus non-parametric datasets. Multiple groups were compared with a two-way ANOVA test; post hoc multiple comparisons were performed with the Holm–Sidak method. Two parametric groups were compared with Student's t-test whereas two non-parametric groups were compared with the Mann–Whitney rank sum test. Finally, cumulative frequency distributions were compared with the two-sample Kolmogorov–Smirnov test. Statistical significance was evaluated at $p \leq 0.05$; for the Kolmogorov–Smirnov test, $p \leq 0.01$ was considered statistically significant.

## Acknowledgements

Research in the Butt lab that contributed to this work was funded by the Medical Research Council (MRC)(MR/K004387/1), Biotechnology and Biological Sciences Research Council (BB/P003796/1), Human Frontiers Science Program Organisation (CDA0023/2008 C), and Brain and Behavior Research Foundation (Narsad; ref 19079). Studentships awarded to FG and AM-S were funded by the Wellcome Trust; PGA was funded by an Imperial College London studentship; CV was funded by an MRC studentship. Funding for equipment came from the Wellcome Trust (089286/Z/09/Z) and OUP John Fell Fund (AV6721-C8000). Work in the Molnár laboratory related to early cortical circuit formation was funded by the MRC (G00900901, MR/N026039/1), Royal Society, and Anatomical Society.

## Additional information

### Funding

| Funder | Grant reference number | Author |
| --- | --- | --- |
| Wellcome Trust | 215199/Z/19/Z | Filippo Ghezzi |
| Wellcome Trust | 086362/Z/08/Z | Andre Marques-Smith |
| Medical Research Council | MR/K004387/1 | Simon JB Butt |
| Human Frontier Science Program | CDA0023/2008-C | Simon JB Butt |
| Brain and Behavior Research Foundation | 19079 | Simon JB Butt |
| Wellcome Trust | 089286/Z/09/Z | Simon JB Butt |

The funders had no role in study design, data collection and interpretation, or the decision to submit the work for publication.

## Author contributions
Filippo Ghezzi, Data curation, Formal analysis, Validation, Investigation, Visualization, Methodology, Writing - original draft, Writing - review and editing; Andre Marques-Smith, Conceptualization, Formal analysis, Investigation, Visualization, Methodology, Writing - review and editing; Paul G Anastasiades, Data curation, Formal analysis, Investigation, Visualization, Methodology, Writing - review and editing; Daniel Lyngholm, Data curation, Formal analysis, Investigation, Visualization, Writing - review and editing; Cristiana Vagnoni, Data curation, Formal analysis, Investigation, Writing - review and editing; Alexandra Rowett, Gokul Parameswaran, Data curation, Formal analysis, Investigation, Visualization; Anna Hoerder-Suabedissen, Yasushi Nakagawa, Resources, Writing - review and editing; Zoltan Molnar, Conceptualization, Resources, Funding acquisition, Writing - review and editing; Simon JB Butt, Conceptualization, Resources, Data curation, Formal analysis, Supervision, Funding acquisition, Validation, Investigation, Visualization, Methodology, Writing - original draft, Project administration, Writing - review and editing

## Author ORCIDs
Filippo Ghezzi https://orcid.org/0000-0003-4538-2578
Andre Marques-Smith https://orcid.org/0000-0001-6879-2858
Daniel Lyngholm https://orcid.org/0000-0002-3708-0249
Anna Hoerder-Suabedissen https://orcid.org/0000-0003-1953-7871
Yasushi Nakagawa https://orcid.org/0000-0003-4876-5718
Zoltan Molnar https://orcid.org/0000-0002-6852-6004
Simon JB Butt https://orcid.org/0000-0002-2399-0102

## Ethics
Animal experimentation: Animal care and experimental procedures were approved by the University of Oxford local ethical review committee and conducted in accordance with UK Home Office personal and project (70/6767; 30/3052; P861F9BB75) licenses under the Animals (Scientific Procedures) 1986 Act.

## Decision letter and Author response
Decision letter https://doi.org/10.7554/eLife.60810.sa1
Author response https://doi.org/10.7554/eLife.60810.sa2

# Additional files

## Supplementary files
• Transparent reporting form

## Data availability
All data generated and analysed during this study are available via the University of Oxford open access data repository (https://ora.ox.ac.uk).

The following dataset was generated:

| Author(s) | Year | Dataset title | Dataset URL | Database and Identifier |
|---|---|---|---|---|
| Ghezzi F, Butt SJ | 2021 | Lpar1 Subplate Data | https://ora.ox.ac.uk | ORA, TBC |

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
