## [Decision Letter]

**Acceptance summary:**

Your revised manuscript elegantly reveals the existence of morphologically and functionally entangled subplate circuits, with important insights for our understanding of how cortical circuits form and convey sensory information. The additional experiments and analyses provided in the revised manuscript significantly reinforce the strength of the conclusions and of the interpretation.

**Decision letter after peer review:**

Thank you for submitting your article "Non-canonical role for Lpar1-EGFP subplate neurons in early postnatal somatosensory cortex" for consideration by *eLife*. Your article has been reviewed by 2 peer reviewers, and the evaluation has been overseen by a Reviewing Editor and John Huguenard as the Senior Editor. The following individual involved in review of your submission has agreed to reveal their identity: Guille López-Bendito (Reviewer #2).

The reviewers have discussed the reviews with one another and the Reviewing Editor has drafted this decision to help you prepare a revised submission.

Both reviewers highlighted the interest and novelty of your study- understanding how functional subplate circuits wire up is important to grasp the interplay between cortical development and incoming sensory information.

As you will see below, both reviewers have highlighted the need to: (1) increase the number of experimental points to reinforce the link between morphology and functional data; (2) to provide more detailed analyses of some data to clarify electrophysiological properties and determine whether Lpar1-egfp subplate cells fall into two groups or rather a continuum; (3) clarify the timeline by providing sufficient n and clarity in the presentation.

*Reviewer #1:*

This is a convincing and important study on the morphological and functional properties of a subpopulation of subplate neurons (SPNs) defined by the expression of Lpar1-EGFP. The authors use a variety of technologies to demonstrate that Lpar1-EGFP SPNs consist of two groups with distinct morphology and synaptic inputs.

(1) Figure 2f: Can the disappearance of fusiform cells be explained by apoptosis? Maybe some electrophysiological data would support this (e.g. less negative resting membrane potential, low input resistance, etc).

2) Figure 2h: The authors have no evidence for direct synaptic connection between fusiform and pyramidal SPN, as can be obtained by paired-recordings. I would put a ? there.

3) It is known that SPNs are also coupled via gap junctions. How did the authors differentiate direct/synaptic responses (e.g. Figure 3) from electrical synapses?

4) In discussion it would be nice to briefly discuss that SP became increasingly more important, from initial "waiting station" (classical papers by Carla Shatz) to plasticity gate and amplifier. The present study nicely shows that fusiform Lpar1-EGFP SPNs may act as amplifiers of thalamic inputs.

5) Figure 7: Nice figure, but how about anti-/orthodromic responses of both SPN populations following electrical thalamic stimulation?

*Reviewer #2:*

The authors provide an important contribution to the current knowledge of subplate neuron diversity and their function in the establishment of cortical circuitry at early stages of development. This is a well-conceived and executed study.

The authors use a combination of distinct approaches including: genetic and morphological techniques, electrical stimulation, laser scanning photostimulation and optogenetics, which strengthens the conclusions of the study. The results are novel as they provide specific circuit and developmental data on a transient layer of the cortex, the subplate, whose function is largely unknown.

1. When the authors say that Lpar1-EGFP neurons represent two morphological subtypes (line 28-29, and discussion line 309) it may be implied that all fusiform and pyramidal SPNs belong to this genetic subtype. This is not the case or at least it is not supported by the data. I suggest rephrasing to: "The Lpar1-EGFP group is comprised by two morphological subtypes" or similar.

2. In general, the association between the electrophysiological results and the morphological neuronal subtypes is based on a very low experimental number. In my opinion, the attempt to match the morphological subtypes is somehow forced and dampens the impact of the electrophysiological results. Furthermore, the n is low to fully support some of the conclusions raised and some quantifications and statistical analyses are missing, as specified below.

3. Lines 118-122. Can the authors provide statistical analysis to the data to fully support their conclusion on the existence of a progressive maturation in the electrophysiological properties of SPNs neurons with age?

4. Lines 130-131. These cells probably are part of a continuum, as it is evidenced in figure 2C. To unbiasedly classify cells into these two types, I would suggest a K-means cluster, and to compare statistically the difference in directionality between the two subtypes.

5. Line 132-136. Are all pyramidal SPNs extending their axons to L1 and all fusiform cells keeping theirs in the SP? The exact numbers should be provided.

6. Line 140. Is the size of the soma smaller in fusiform cells? If so, this might generate a bias when patching and filling the cells, thus giving the wrong impression that their numbers are decreased. Increasing the n at later ages would probably help. Can the apical process of pyramidal SPNs be detected from the Lpar1-EGFP sections directly? If this is the case, counting a higher number of cells in several sections from different brains might provide a better estimate of the population and thus of the reduction in number of fusiform cells. This is an important point in this paper and a key element of the circuit maturation proposed in this work.

7. Line 144-Figure 2G. Can the authors provide a quantification? (At least n and percentage)

8. Line 178-180. Can the authors specify at which ages these cells morphologies were recovered or whether they appear at all ages studied? In line 180 it says 5/5 pyramidal SPNs and in line 181 it says 6/9 pyramidal. This whole paragraph is confusing and the conclusion is important. Again, the numbers are too small to make a strong association between morphology and input.

9. Line 199-207. To strengthen the conclusion that there is a switch in the distribution of inputs then should be comparable for both the P1-4 and the P5-8 groups.

10. Line 191-214. In this paragraph the association between morphology and local/translaminar input is not made. Explicitly say why.

11. Line 212-213 "local GABAergic input were primarily pyramidal subtypes". Please provide N.

12. Line 214. Do you mean n=4 out of 4?

13. Line 247. Why is this data not shown?

14. Line 250. "…precedes our previously reported reciprocal connection between these SST+ cells and L4 spiny stellate neurons" At what age is this? Can you include it in the text (in addition to giving the reference)?

15. Line 262-263. Is this "drop in incidence" statistically significant?

16. Line 298. Again, providing morphological data with N=1 does not add up but rather distracts from the conclusions.

17. Line 300. How many non-EGFP SPNs have been recorded? Again, can the authors provide a statistical test to show that connectivity in the Lpar1 population is significantly lower than in the general SPN population?

Figures

18. Figure 1A. The magnification is too low to see the recorded cell(s), please provide a better image or add an inset of the cell(s).

19. Figure 2H. The diagram might be misleading since it gives the impression that fusiform cells target specifically Lpar1-pyramidal SPNs and the authors just demonstrate that the axons are restricted to the subplate. Thus, their target may or may not include, and may or may not be restricted to Lpar1-pyramidal SPNs.

20. Figure 3D. Why is the resolution in the left panel different from that in 3B or 3F?

21. Figure 5: Why is P5-8 data missing in this condition? Can the authors include it and make a figure with a layout consistent with figure 4?

22. Figure 5A-C. N is not specified.

23. Fig5F Please note that this panel is not referenced in the text.

24. Fig7. Legend on the figure would be helpful.

---

## [Author Response]

Reviewer #1:[…] (1) Figure 2f: Can the disappearance of fusiform cells be explained by apoptosis? Maybe some electrophysiological data would support this (e.g. less negative resting membrane potential, low input resistance, etc).

We agree with the reviewer that this is a possible explanation. We revisited our intrinsic electrophysiology data but found no clear evidence of markers for apoptosis. This is likely in part due to the variability in the dataset and the fact that relatively few *Lpar1-EGFP* SPNs undergo cell death. Towards addressing this point we do provide new data from immunohistochemistry experiments in which we assessed the presence of cell death markers at the P3-4 to P5-6 juncture (Figure 4 and accompanying text).

2) Figure 2h: The authors have no evidence for direct synaptic connection between fusiform and pyramidal SPN, as can be obtained by paired-recordings. I would put a ? there.

The reviewer is correct, there is no evidence to support a direct synaptic connection between the two *Lpar1-EGFP* subtypes. One limitation of our study (see discussion) is that we cannot map the output of *Lpar1-EGFP* SPNs and are reliant on recovered morphologies to determine the location of putative efferent targets. We have amended Figure 2h such that (i) the axon of the fusiform cell extends beyond the pyramidal (Pyr) subtype, with (ii) a collateral (marked with a question mark) engaging with the Pyr subtype as suggested by the reviewer, and further explained in the figure legend.

3) It is known that SPNs are also coupled via gap junctions. How did the authors differentiate direct/synaptic responses (e.g. Figure 3) from electrical synapses?

We agree that gap junction coupled spikes can at times appear like short, sharp synaptic events. Three observations suggest however that our events are due to chemical neurotransmission: (1) the latency of synaptic events likely – but not absolutely – precludes the fact that these are action potentials in a gap junction coupled cell; (2) gap junction coupling invariably acts as a low pass filter and we observed no slow outward current reflecting the after-hyperpolarisation of the action potential. Moreover, the variable amplitude, kinetics and summation of the synaptic potentials reflect those associated with chemical neurotransmission; (3) the amplitude of the synaptic response varied with the holding potential. While we did not explicitly test the contribution of gap junction coupling, for example with pharmacological blockers, we are confident that the observed response are chemical in nature.

4) In discussion it would be nice to briefly discuss that SP became increasingly more important, from initial "waiting station" (classical papers by Carla Shatz) to plasticity gate and amplifier. The present study nicely shows that fusiform Lpar1-EGFP SPNs may act as amplifiers of thalamic inputs.

We welcome the reviewer’s input and have added text to this effect. This reflects increasing evidence from a number of labs that is transforming the way we view the function of subplate in perinatal mammalian neocortex.

5) Figure 7: Nice figure, but how about anti-/orthodromic responses of both SPN populations following electrical thalamic stimulation?

We thank the reviewer for the positive comments on the figure. We observed no antidromic action potentials in *Lpar1-EGFP* SPNs in response to electrical stimulation of the thalamus (now stated in the section detailing our thalamic electrical stimulation experiments). This suggests that these neurons are unlikely to be corticothalamic, although we cannot entirely discount that such connections are severed in preparation of the acute in vitro thalamocortical slices. We have reworded the text at the end of these results to further clarify that the majority of EPSCs evoked by electrical stimulation are likely via antidromic activation of corticothalamic (CT) projection neurons. We considered including this neuron in Figure 7b but decided against this as we neither know the location of these CT neurons in infragranular layers or how they integrate into the wider circuit at these early ages.

Reviewer #2:[…] 1. When the authors say that Lpar1-EGFP neurons represent two morphological subtypes (line 28-29, and discussion line 309) it may be implied that all fusiform and pyramidal SPNs belong to this genetic subtype. This is not the case or at least it is not supported by the data. I suggest rephrasing to: "The Lpar1-EGFP group is comprised by two morphological subtypes" or similar.

We completely agree with the reviewer and have amended the text accordingly

2. In general, the association between the electrophysiological results and the morphological neuronal subtypes is based on a very low experimental number. In my opinion, the attempt to match the morphological subtypes is somehow forced and dampens the impact of the electrophysiological results. Furthermore, the n is low to fully support some of the conclusions raised and some quantifications and statistical analyses are missing, as specified below.

We appreciate the concern of the reviewer in the relatively small number of n-numbers and associated recovered morphologies for some of the experiments. Unfortunately recovering morphologies at such early stages is problematic due in part to the small size and fragile nature of the neurons at these time points. This is further compounded by the fact that during long duration experiments, such as LSPS recordings, the neurons end up being tightly adhered to the patch electrode such that they are invariably pulled out of the tissue on withdrawal of the electrode at the end of the experiment. To address this concern we have made modifications to the text and included additional data. Overall these support our claim that P5 is a critical juncture in the *Lpar1-EGFP* SPN circuit with fusiform subtypes – that receive translaminar input, largely absent from our sample from this time point onward.

3. Lines 118-122. Can the authors provide statistical analysis to the data to fully support their conclusion on the existence of a progressive maturation in the electrophysiological properties of SPNs neurons with age?

We have provided this analysis as requested as we agree that this is important. We excluded it from the initial submission to avoid over-complicating Figure 1 and due to the fact that the trends shown by the box plots were fairly self-evident. Multiple comparisons between the groups revealed significant differences in 5 out of 6 intrinsic electrophysiological properties; the only exception is resting membrane potential. The outcome of the statistical tests are reported in Figure 1 legend and briefly explained in the first paragraph of the Results section .

4. Lines 130-131. These cells probably are part of a continuum, as it is evidenced in figure 2C. To unbiasedly classify cells into these two types, I would suggest a K-means cluster, and to compare statistically the difference in directionality between the two subtypes.

We thank the reviewer for this comment. The original text was poorly worded and as such we have reworded the section dealing with figure 2a-c to emphasise that the key difference between the two subtypes in terms of dendritic arbour was the presence or absence of an apical dendrite. The directionality is interesting as it confirms that while some pyramidal subtypes have apical dendrites that are tilted (for example the cell shown in figure 2a,b), they are still – in their overall dendritic arbour – distinct from bitufted fusiform cells; a subtype of Lpar1-EGFP+ neuron that are dominated by dendrites running in parallel with the subplate/white matter tract.

5. Line 132-136. Are all pyramidal SPNs extending their axons to L1 and all fusiform cells keeping theirs in the SP? The exact numbers should be provided.

The text detailing our analysis of the 19 morphologies has been amended to address the comment of the reviewer . All fusiform cells (7/7) with complete or near complete morphologies extended laterally in the SP/L6a/white matter tract. For the majority (9/12) of pyramidal cells we could trace axons projecting to L1. The axons of the remaining 3 extend to L2 where they appeared to be severed. As mentioned in the text, pyramidal subtypes also had collaterals – sometimes quite extensive – in the subplate as evident in figure 2d.

6. Line 140. Is the size of the soma smaller in fusiform cells? If so, this might generate a bias when patching and filling the cells, thus giving the wrong impression that their numbers are decreased. Increasing the n at later ages would probably help. Can the apical process of pyramidal SPNs be detected from the Lpar1-EGFP sections directly? If this is the case, counting a higher number of cells in several sections from different brains might provide a better estimate of the population and thus of the reduction in number of fusiform cells. This is an important point in this paper and a key element of the circuit maturation proposed in this work.

There is no discernible difference in the size of the soma of the two subtypes. Our observed increase in the proportion of pyramidal subplate cells is consistent with previous reports (Marx et al., 2017). That said, we were (similar to the reviewer) surprised to recover no fusiform *Lpar1-EGFP* SPNs at later ages. Unfortunately we have been unable to discern the presence or absence of the apical dendrite in *Lpar1-EGFP* section due to the dense nature of the processes, and the fact that the pyramidal apical dendrites do not extend beyond L6a. We agree that the disappearance and possible cell death of the fusiform subtype is an important part of the paper. As such, we have carried out a detailed assessment of cell death at the P3-4 to P5-6 transition and provide evidence that suggests that a subset of *Lpar1-EGFP* subplate cells do undergo cell death at this time point (Figure 4 and associated text ). This supports the idea that this time window represents an important transition in the early circuits of S1BF.

7. Line 144-Figure 2G. Can the authors provide a quantification? (At least n and percentage)

Counts from n=3 animals are now provided on lines 179-80. None of the 79 EGFP+ SPNs counted across images of subplate obtained from these 3 animals co-expressed GABA. Whereas, 16/19 Lpar1-EGFP+ cells in L5 (84%) co-expressed GABA, in agreement with our previous report (Marques-Smith et al., 2016).

8. Line 178-180. Can the authors specify at which ages these cells morphologies were recovered or whether they appear at all ages studied? In line 180 it says 5/5 pyramidal SPNs and in line 181 it says 6/9 pyramidal. This whole paragraph is confusing and the conclusion is important. Again, the numbers are too small to make a strong association between morphology and input.

We agree with reviewer that the text was confusing and have reworded the relevant section to more clearly state the number of morphologies recovered during these LSPS experiments. The numbers recovered represent 38% (15/40) of patched neurons from P1-4 and 35% of those recorded between P5 and P8. These numbers are comparable with the gold standard morphological assessment of subplate/L6b neurons reported by Marx and colleagues (2017)(Cerebral Cortex 27: 1011–1026): they analysed 76 of 230 recorded neurons (33%). Moreover, previous LSPS papers on subplate connectivity report no recovered morphologies (for example Viswanathan et al. (2012) J. Neurosci. 32 (5) 1589-1601). It is worth reiterating that recovering complete morphologies at these early ages is challenging especially after prolonged recording. Further, we believe that inclusion of these data is merited as it provides further insight into the classification and diversity of these cells through this critical first week of development.

9. Line 199-207. To strengthen the conclusion that there is a switch in the distribution of inputs then should be comparable for both the P1-4 and the P5-8 groups.

To address the reviewer’s concern we have performed additional LSPS recordings of GABAergic input onto SPNs in P1-P4 animals. Of the 5 cells mapped, 4 were characterized by local GABAergic inputs and 1 received translaminar input from L5. These data have been incorporated into the P1-4 data shown in Figure 5c-e and 5i. We have also run a chi-square test on the proportion of translaminar and local cells across the two developmental time points. The proportion of the two subtypes does not differ significantly (chi-square = 0.533; p = 0.46). However, it is clear from the summary data that there is a shift from localised input from L5 (Figure 5d,e) to diverse inputs that on average represent a diffuse columnar input at P5-8 (Figure 5g,h).

10. Line 191-214. In this paragraph the association between morphology and local/translaminar input is not made. Explicitly say why.

We have reworded the text at the end of this section (lines 290-294) to explicitly state the number of recovered morphologies.

11. Line 212-213 "local GABAergic input were primarily pyramidal subtypes". Please provide N.

We have altered this text to include n-numbers for all data as requested.

12. Line 214. Do you mean n=4 out of 4?

Yes, this has been reworded but we only recovered 4 morphologies from the 16 cells recorded from P5-8 (25%); all were of the pyramidal subtype.

13. Line 247. Why is this data not shown?

The rationale for the P2X2 optogenetic actuator experiments was simply to confirm if the source of the early translaminar input was indeed L5 SST+ interneurons. ChR2 optogenetic experiments – shown in Figure 6a-d, do not reveal the layer location of the presynaptic SST+ cells. We have now reworded this section to emphasise our focus on the P1-P4 time window where we demonstrate that the translaminar input originates from SST+ INs.

14. Line 250. "…precedes our previously reported reciprocal connection between these SST+ cells and L4 spiny stellate neurons" At what age is this? Can you include it in the text (in addition to giving the reference)?

We thank the reviewer for this spotting this oversight on our part. We have previously reported a translaminar input from L5 SST+ interneurons onto L4 SSNs between P4-P9 – now included in the text along with the associated reference.

15. Line 262-263. Is this "drop in incidence" statistically significant?

We performed a chi-square test comparing P1-4 versus P5-8 (Χ2 (1, N = 56) = 5.364, p = 0.021). The significant drop at the later ages is reported line 403.

16. Line 298. Again, providing morphological data with N=1 does not add up but rather distracts from the conclusions.

We agree with the reviewer. It is startling that we only find 2 out of 19 cells with confirmed thalamic input. While it is fortunate that we recovered both morphologies, we appreciate that n=1 is distracting from this important message and have – in line with the recommendation of the reviewer – deleted this text.

17. Line 300. How many non-EGFP SPNs have been recorded? Again, can the authors provide a statistical test to show that connectivity in the Lpar1 population is significantly lower than in the general SPN population?

In the original submission we reported n=5 non-EGFP SPNs. We have now recorded a further 10 cells to bring this number up to 15 of which 3 received confirmed thalamic input. This relatively low percentage (20%) is similar to that observed with *Lpar1-EGFP* SPNs (11%) and distinct from previous reports that have relied on electrical stimulation alone.

Figures18. Figure 1A. The magnification is too low to see the recorded cell(s), please provide a better image or add an inset of the cell(s).

The purpose of this image was to make clear the located of the subplate relative to full depth of cortex in S1BF. Detailed morphologies are provided in figure 2.

19. Figure 2H. The diagram might be misleading since it gives the impression that fusiform cells target specifically Lpar1-pyramidal SPNs and the authors just demonstrate that the axons are restricted to the subplate. Thus, their target may or may not include, and may or may not be restricted to Lpar1-pyramidal SPNs.

We agree with both reviewers on this point and have reviewer 1 recommended (major comment #2) amended Figure 2h.

20. Figure 3D. Why is the resolution in the left panel different from that in 3B or 3F?

We apologise for this error. On occasion conversion of the Matlab figure into PDF or equivalent format causes a loss of resolution. We have amended and will double check the submitted version.

21. Figure 5: Why is P5-8 data missing in this condition? Can the authors include it and make a figure with a layout consistent with figure 4?

See our comment above – the purpose of the P2X2 optogenetic actuator experiments was to assess if the transient translaminar input from L5 – present from P1 to P4 – originated from SST+ interneurons.

22. Figure 5A-C. N is not specified.

Apologies for this oversight. We have now included the N numbers in the figure legend.

23. Fig5F Please note that this panel is not referenced in the text.

Thank you for pointing this out! Figure 5f (*now* figure 6f) is now referred to in the text.

24. Fig7. Legend on the figure would be helpful.

Thank you for the suggestion: we have added a legend.